# A photo-triggered self-accelerated nanoplatform for multifunctional image-guided combination cancer immunotherapy

Xiaoying Kang[1,4], Yuan Zhang[1,4], Jianwen Song[1], Lu Wang[2], Wen Li ⊕[2] ✉, Ji Qi ⊕[1] ✉ & Ben Zhong Tang ⊕[3] ✉

Precise and efficient image-guided immunotherapy holds great promise for cancer treatment. Here, we report a self-accelerated nanoplatform combining an aggregation-induced emission luminogen (AIEgen) and a hypoxia-responsive prodrug for multifunctional image-guided combination immunotherapy. The near-infrared AIEgen with methoxy substitution simultaneously possesses boosted fluorescence and photoacoustic (PA) brightness for the strong light absorption ability, as well as amplified type I and type II photodynamic therapy (PDT) properties via enhanced intersystem crossing process. By formulating the high-performance AIEgen with a hypoxia-responsive paclitaxel (PTX) prodrug into nanoparticles, and further camouflaging with macrophage cell membrane, a tumor-targeting theranostic agent is built. The integration of fluorescence and PA imaging helps to delineate tumor site sensitively, providing accurate guidance for tumor treatment. The light-induced PDT effect could consume the local oxygen and lead to severer hypoxia, accelerating the release of PTX drug. As a result, the combination of PDT and PTX chemotherapy induces immunogenic cancer cell death, which could not only elicit strong antitumor immunity to suppress the primary tumor, but also inhibit the growth of distant tumor in 4T1 tumor-bearing female mice. Here, we report a strategy to develop theranostic agents via rational molecular design for boosting antitumor immunotherapy.

Precision medicine calls for efficient strategies that enable comprehensive diagnosis and therapeutics of diseases[1,2]. However, most of the currently available techniques face some shortcomings and are hindered by the unsatisfied outcome. Fluorescence imaging possesses high sensitivity, but the tissue penetration ability is limited[3–5]. In contrast, photoacoustic (PA) imaging has good penetration depth and spatial resolution, but the sensitivity is relatively low[6–9]. Therefore, the integration of fluorescence and PA imaging could provide comprehensive information about diseases. However, since the fluorescence and PA signals are respectively related to the radiative and non-radiative channels of a chromophore, it is difficult to obtain maximal brightness of them at the same time[10–13]. On the other hand, traditional first-line cancer treatment methods such as surgery, chemotherapy, and radiotherapy usually have limited antitumor outcomes due to the

[1]State Key Laboratory of Medicinal Chemical Biology, Frontiers Science Center for Cell Responses, Key Laboratory of Bioactive Materials, Ministry of Education, and College of Life Sciences, Nankai University, Tianjin 300071, China. [2]Tianjin Key Laboratory of Biomedical Materials and Key Laboratory of Biomaterials and Nanotechnology for Cancer Immunotherapy, Institute of Biomedical Engineering, Chinese Academy of Medical Sciences and Peking Union Medical College, Tianjin 300192, China. [3]School of Science and Engineering, Shenzhen Institute of Aggregate Science and Technology, The Chinese University of Hong Kong, Shenzhen 518172 Guangdong, China. [4]These authors contributed equally: Xiaoying Kang, Yuan Zhang. ✉e-mail: liwen@bme.pumc.edu.cn; qiji@nankai.edu.cn; tangbenz@cuhk.edu.cn

tumor recurrence/metastasis. Recently, immunotherapy has changed the paradigm of cancer therapy, and shows great potential to cure malignant tumors[14–16]. However, the clinical application of immunotherapy still faces the problem that a majority of solid tumors have low immunogenicity, and do not respond to immunotherapy, which are defined as "cold tumors"[17–21].

Immunogenic cell death (ICD) is a promising approach to activate the immunogenicity of tumor cells for enhanced antitumor immune responses, during which tumor cells secrete immunostimulatory damage-associated molecular patterns (DAMPs) including surface-exposed calreticulin (ecto-CRT), high mobility group protein B1 (HMGB1), heat shock protein 70 (HSP70), and adenosine triphosphate (ATP) to promote T cells infiltration and transform immunologically cold tumors to hot[22–26]. Recently, it has been demonstrated that photodynamic therapy (PDT), a process that a photosensitizer reacts with oxygen to produce singlet oxygen ($^1O_2$) and/or reactive oxygen species (ROS) under light irradiation, could not only destroy tumor cells directly but also elicit immunogenicity by effectively inducing ICD[27–31]. However, the PDT-based ICD inducers still face several obstacles. First, conventional organic dyes including the clinically used methylene blue (MB) and indocyanine green (ICG) suffer from the aggregation-caused quenching (ACQ) effect, which significantly depresses the fluorescence emission and PDT property[32,33]. Aggregation-induced emission (AIE) emerges as a new photophysical phenomenon that could solve the ACQ problem and enable high-performance emitters and photosensitizers[34–37]. Nevertheless, some AIE luminogens (AIEgens) suffer from the issue of relatively weak light absorption ability, in which strong light irradiation has to be used to trigger efficient PDT[38,39]. Second, due to the rapid growth rate and oxygen consumption of tumor cells, hypoxia is a conspicuous feature in the microenvironment of many malignant solid tumors, which, however, declines the performance of PDT for treating solid tumor[40,41]. In addition, the limited penetration depth of light is also an impact factor of phototherapy including PDT. Light can hardly penetrate into deep tumor tissue, which depresses the antitumor effect. Therefore, the integration of phototherapy with other therapeutic modalities would be a good choice to boost cancer immunotherapy outcome[42,43].

Some chemotherapeutic drugs (e.g., doxorubicin (DOX) and paclitaxel (PTX)) have been reported to possess immune-regulation ability[44–46]. For example, PTX can act as an inducer of endogenous vaccines through ICD effect. Nevertheless, small molecule drugs would cause systemic toxicity due to the poor targeting ability to disease site. The drug delivery system based on nanotechnology can promote drug accumulation in diseases. Despite the nanodrug delivery system could improve the accumulation of chemotherapeutic drug at tumor site via enhanced permeability and retention (EPR) effect, it would also move to other organs especially the reticuloendothelial system (RES) such as liver and spleen and cause burden and even damage to these organs. A much safer way is to make use of prodrugs that are safe and will release free drug upon specific external trigger or in the disease microenvironment[47–49]. The hypoxia nature is a critical marker in the tumor microenvironment (TME), which could be utilized to design hypoxia-responsive prodrugs that can release drugs at the tumor site[50,51]. However, the hypoxic TME is usually not enough to trigger the prodrug conversion rapidly.

In this work, we develop a kind of organic phototheranostic platform for self-accelerated multifunctional image-guided combination immunotherapy (Fig. 1). To obtain a high-performance phototheranostic agent, two near-infrared (NIR) AIEgens with different substitutes are designed and synthesized. By systematic comparison of the properties, it is found that the introduction of methoxy group is able to simultaneously increase the fluorescence and PA brightness for the strong light absorption ability. Noteworthy, both type I and type II ROS generation properties are amplified through the more efficient intersystem crossing (ISC) process. By combining the

high-performance AIEgen with PTX-based hypoxia-responsive prodrug into nanoparticles (NPs), and further camouflaging with M1 macrophage cell membrane, a tumor-targeting theranostic agent is designed, which exhibits enhanced accumulation at the tumor site. For their complementary advantages, in vivo fluorescence and PA imaging help to delineate tumor regions sensitively, providing accurate guidance for tumor therapy. The light-induced ROS production is capable of inducing the ICD of tumor cells, which could integrate with hypoxia-triggered PTX release to improve the antitumor immune responses. It is noted that the PDT effect could consume the local oxygen and lead to more severely hypoxic TME, accelerating the release of hypoxia-responsive prodrug. As a result, the self-synergistic immunotherapy of PDT and PTX drug could not only elicit strong antitumor immunity to suppress the primary tumor, but also inhibit the growth of distant tumor. By combining the high-performance phototheranostic AIEgen and hypoxia-responsive prodrug, a self-accelerated nanotheranostic system is constructed for multifunctional image-guided self-synergistic immunotherapy. This work present a strategy to develop robust theranostic agents for boosting antitumor immunotherapy.

## Results

### Synthesis and characterization of AIEgens

Two donor-acceptor-donor (D-A-D) type compounds with different substitutes were first designed and synthesized to obtain low-bandgap chromophores. Herein, tetraphenyl ethylene (TPE) or methoxy-substituted TPE (MTPE) and thieno[3,4-c][1,2,5]thiadiazole (TT) were employed as the D and A moieties, respectively. TPE is a widely used building block for constructing AIEgens, and MTPE is a derivative with stronger electron-donating property. TT is an analog of the popularly used electron acceptor benzothiadiazole, but TT is expected to have much stronger electron-withdrawing ability for the quinoid resonance structure and supervalence sulfur atom[52,53]. The molecules were synthesized via Suzuki cross-coupling reaction between the mono boric acid ester of TPE or MTPE and 2,5-dibromo-3,4-dinitrothiophene to produce the dinitro compound, followed by reduction to the diamino intermediate, and subsequently underwent ring-closing reaction with N-sulfinylaniline to yield the target molecule (Fig. 2a, Supplementary Figs. 1 and 2). The intermediates and final compounds have been fully characterized by nuclear magnetic resonance (NMR) and high-resolution mass spectra (HRMS) (Supplementary Figs. 3–22). TPE-TT and MTPE-TT had good processability and they could be easily dissolved in common organic solvents such as toluene, dichloromethane, and tetrahydrofuran (THF).

Density function theory (DFT) calculation was conducted to gain in-depth understanding about the molecular structure and electronic property[54]. As shown in Fig. 2b, these two molecules possess similar geometry. The core unit (Ph-TT-Ph) is a relatively planar structure with very small dihedral angles (<5°) between TT and the neighboring phenyl rings, in which the excellent conjugation is advantageous for efficient intramolecular charge transfer (ICT) between D and A units, and strong absorption ability as well. Other phenyl rings in TPE or MTPE have highly twisted conformation with crowded 3D geometry, which would be favorable for realizing AIE feature. Moreover, the strong intramolecular motion of phenyl rotors is expected to boost the PA effect[55,56]. The electron cloud of the lowest unoccupied molecular orbital (LUMO) mainly locates in the electron-withdrawing TT group, while the highest occupied molecular orbital (HOMO) distributes along the whole molecule, which suggests an efficient ICT effect. From TPE to the more electron-donating MTPE, the LUMO energy level nearly doesn't change, while the HOMO increases from −4.94 eV to −4.79 eV, which results in decreased energy bandgap and longer response wavelength.

### Photophysical property

As presented in Supplementary Fig. 23, the two molecules had similar absorption profiles in THF, and the maximal absorption wavelength of

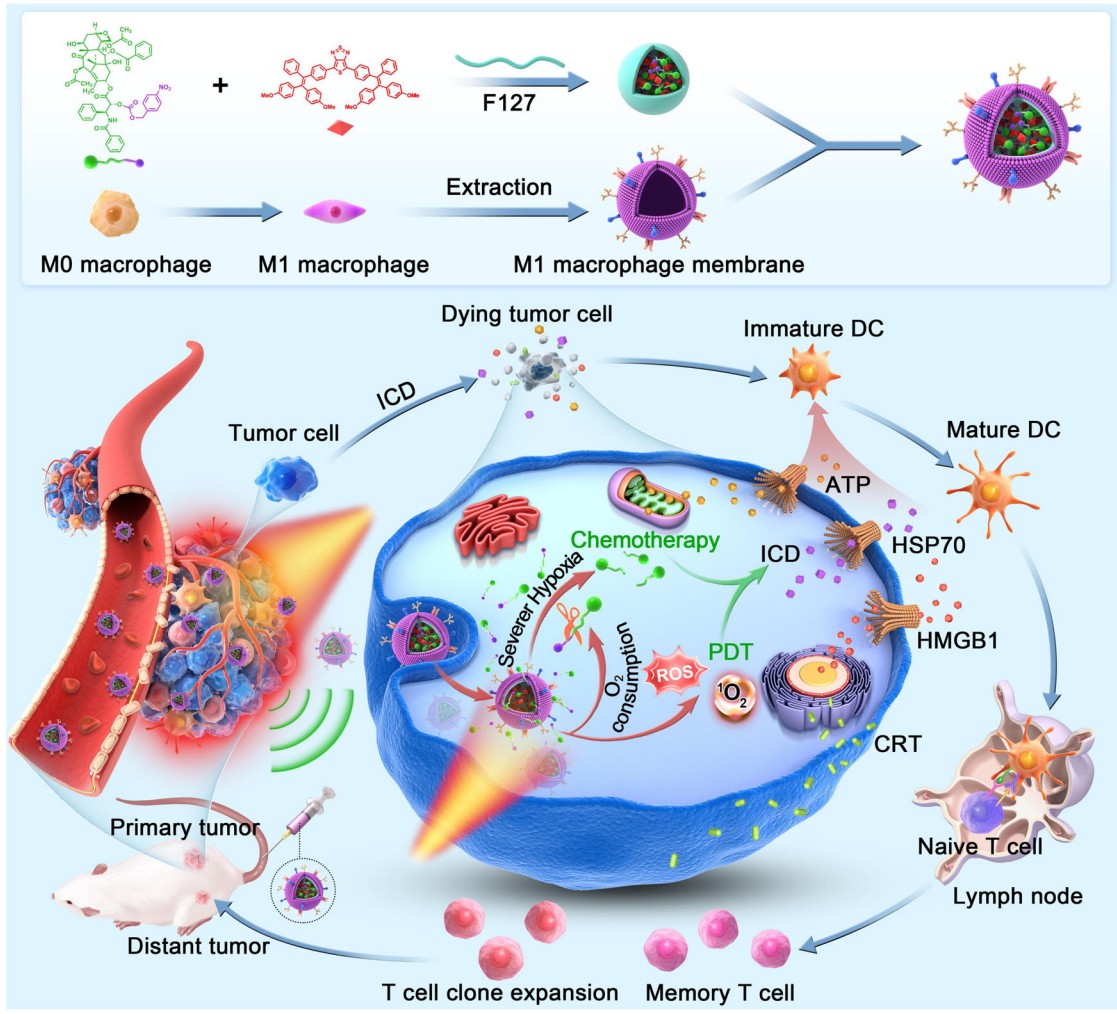

**Fig. 1 | Schematic illustration of the photo-triggered self-accelerated nano-platform for fluorescence and PA imaging-guided combination immunotherapy.** In this work, an AIEgen with bright NIR fluorescence and PA signal and both type I and type II PDT properties was synthesized. By co-assembly of the AIEgen and a hypoxia-responsive PTX prodrug with Pluronic F-127, a theranostic nanoplatform was built, which was further coated with M1 macrophage cell membrane to endow tumor targeting ability. The PDT effect could not only induce ICD but also accelerate the activation of hypoxia-responsive prodrug by consuming the local oxygen in tumor site. As a result, the combination of PDT and self-accelerated PTX release elicited potent immune response and excellent antitumor outcomes.

MTPE-TT (624 nm) was red-shifted as compared with TPE-TT (609 nm) due to the reduced bandgap from stronger D-A interaction. Note-worthy, the maximal molar absorption coefficient of MTPE-TT was $2.49 \times 10^4 \, M^{-1} \, cm^{-1}$, much higher than that of TPE-TT ($1.85 \times 10^4 \, M^{-1} \, cm^{-1}$), being beneficial for efficient light excitation. To study the fluorescence property in different aggregate states, the photoluminescence (PL) spectra in THF/water mixture with various water fractions ($f_w$) were recorded. As shown in Fig. 2c, d and Supplementary Fig. 24, the PL intensity of both MTPE-TT and TPE-TT decreased when $f_w$ increased to 60% or 70% for the formation of twisted ICT (TICT) state in high polarity environment. TPE-TT then underwent slight PL intensity enhancement when further increasing $f_w$, while MTPE-TT showed very pronounced PL amplification at higher $f_w$ due to the formation of aggregate. As a consequence, MTPE-TT possessed potent AIE signature, and TPE-TT only exhibited weak AIE phenomenon. We further investi-gated the fluorescence property of these molecules in varied viscosity environments. By increasing the glycerol fraction ($f_g$) in the mixture of $N,N$-dimethylformamide (DMF)/glycerol, the fluorescence intensity decreased at first, and then intensified, which could also be assigned to the polarity change and AIE signature (Supplementary Fig. 25). The PL amplitude increase of MTPE-TT at 90% of $f_g$ was about 8-fold higher than that of TPE-TT, agreeing well with the PL change in THF/water mixture, and further confirming the more obvious AIE characteristic of MTPE-TT.

In order to obtain biocompatible agents, the hydrophobic com-pounds were encapsulated into water-soluble NPs with nanoprecipi-tation method using Pluronic F127 as the surfactant. As shown in Fig. 2e, f, the NPs showed similar absorption and emission spectra as that of the THF solution, with a slight bathochromic shift of the absorption spectra and a redshift of the PL spectra were observed. The PL quantum yield (PLQY) of TPE-TT NPs (TNPs) and MTPE-TT NPs (MNPs) were measured to be 4.2% and 7.3%, respectively. According to the energy gap law, the fluorescence brightness usually decreases as the wavelength redshifts, which is more pronounced for NIR emitters[57]. In this work, although MTPE-TT possessed a smaller bandgap, its PLQY also intensified thanks to the prominent AIE feature. Then we investi-gated the ROS generation property. 9,10-Anthracenediyl-bis(methy-lene)dimalonic acid (ABDA) was used as the indicator of $^1O_2$ and 2′,7′-dichlorodihydrofluorescein diacetate (DCF-DA) was used to detect ROS[58,59]. Upon white light irradiation, MNPs showed much stronger $^1O_2$ and ROS generation ability than TNPs (Fig. 2g, h), which also revealed that both type I and type II PDT processes occurred in this NIR lumi-nogen. To uncover the underlying mechanism of this favorable PDT effect, the calculation of energy levels was conducted. As shown in

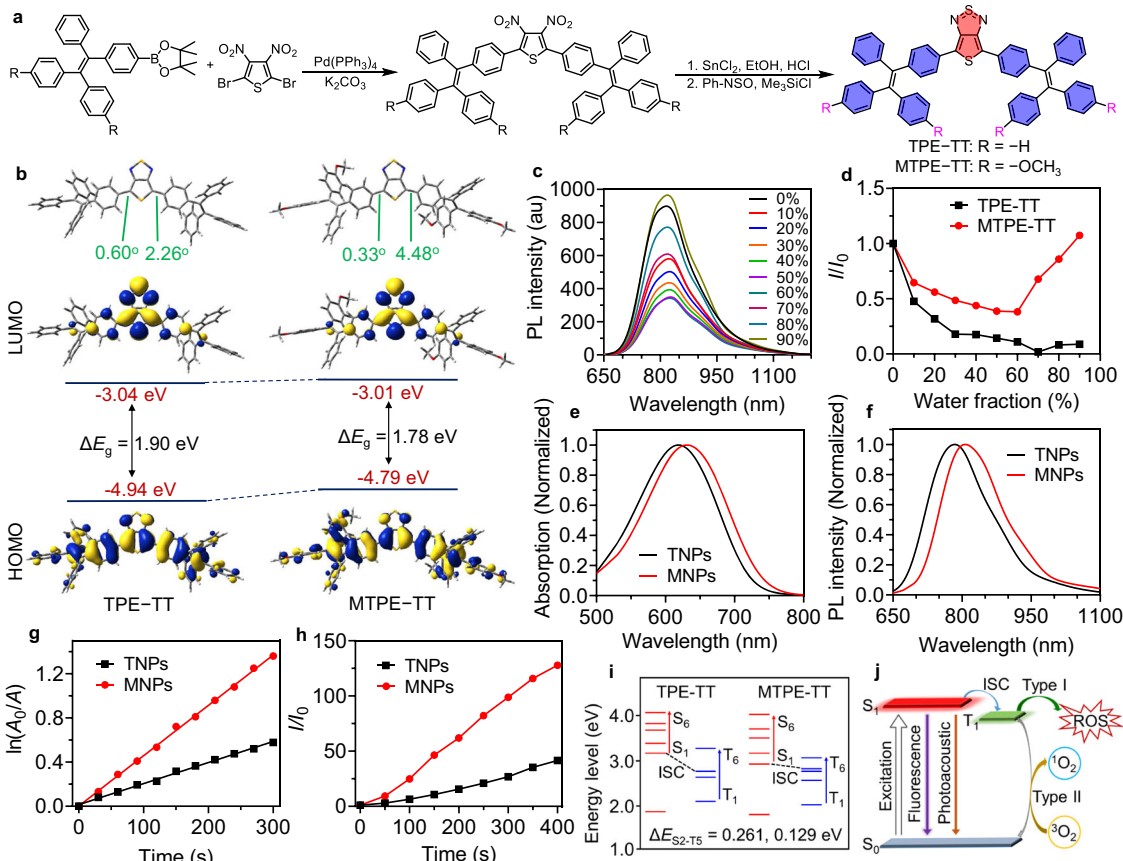

**Fig. 2 | Photophysical property of molecules and NPs. a** Synthetic route to TPE-TT and MTPE-TT. **b** The calculated molecular geometry, HOMO, LUMO, and energy levels of TPE-TT and MTPE-TT. **c** PL spectra of MTPE-TT in THF/water mixture with various water fractions. **d** Plots of the PL peak intensity versus water fraction in THF/water mixture, where $I_0$ and $I$ are the PL intensity in pure THF and THF/water mixtures with various water fractions, respectively. **e**, **f** Absorption and PL spectra of TNPs and MNPs. **g** Plot of $\ln(A_0/A)$ of ABDA versus light irradiation time, where $A_0$ and $A$ are the absorbance of ABDA (378 nm) in the presence of TNPs or MNPs before and after light irradiation, respectively. **h** Plots of $I/I_0$ of DCF-DA versus light irradiation time, where $I_0$ and $I$ are the PL intensity of DCF-DA at 525 nm in the presence of TNPs or MNPs before and after light irradiation, respectively. **i** Schematic illustration of the energy levels and ISC transition of TPE-TT and MTPE-TT. **j** Schematic illustration of the photoexcitation energy transformation processes of MTPE-TT. For **c**–**h**, experiment was repeated three times independently with similar results. Source data are provided as a Source Data file.

Fig. 2i, the lowest energy gap ($\Delta E_{ST}$) between the singlet excited state (S) and triplet excited state (T) of the methoxy-containing MTPE-TT (0.129 eV) was much smaller than that of TPE-TT (0.261 eV), suggesting more efficient ISC process and being responsible for the excellent ROS production capability. We further measured the PA spectra of the NPs solution, and the PA amplitude of MNPs was stronger than that of TNPs (Supplementary Fig. 26). The possible reason was that the higher molar absorption coefficient of MTPE-TT resulted in much stronger light absorption. As schematically illustrated in Fig. 2j, the TT-based AIEgens possessed bright fluorescence and PA imaging property, as well as type I and type II PDT effects, being favorable for multifunctional image-guided therapy applications. More interestingly, as compared with TPE-TT analog, MTPE-TT exhibited red-shifted absorption/emission wavelength, increased fluorescence and PA brightness, and boosted light-triggered ROS generation ability, which suggested that the methoxy substitution of TPE represented an effective strategy to construct highly efficient phototheranostic agent.

### Characterization and hypoxia-responsive nature of prodrug

To obtain a hypoxia-responsive prodrug, the chemotherapeutic drug PTX was conjugated with 4-nitrobenzyl chloroformate to afford PTX-NB[60]. As displayed in Fig. 3a, PTX-NB was expected to convert into free active drug in hypoxic environment following the hypoxic bioreduction to yield the unstable amine intermediate, which would undergo spontaneous self-elimination to release PTX. The synthesis and

characterization of PTX-NB prodrug were depicted in Supplementary information. The ¹H NMR spectra of PTX and PTX-NB suggested the correct chemical structure of the prodrug (Fig. 3b). HRMS indicated a peak at $m/z$ 1071.3365 (Fig. 3c), which was consistent with "PTX-NB + K⁺", and confirmed the successful conjugation of nitrobenzene and PTX. In order to investigate the hypoxic responsiveness of the prodrug, PTX-NB was reacted with different concentrations of sodium dithionite ($Na_2S_2O_4$) solution that was used to mimic the hypoxic condition, and the reaction system was monitored by high-performance liquid chromatography (HPLC). As displayed in Fig. 3d, with the increase of $Na_2S_2O_4$ concentration, the peak of PTX-NB at 10.2 min gradually decreased while a new peak representing PTX at 6.4 min appeared and increased. When the concentration of $Na_2S_2O_4$ was 25 mM, the peak at 10.2 min disappeared and the prodrug was completely converted to free PTX. In addition, after reacting with $Na_2S_2O_4$, the product was measured by HRMS, the peak at $m/z$ 876.3205 was consistent with "PTX + Na⁺" (Fig. 3e). These results revealed the hypoxia-triggered degradation nature of PTX-NB and the controlled release of PTX.

### Characterization of M1 macrophage membrane-camouflaged NPs

After confirming the excellent photophysical and PDT properties of MTPE-TT and the hypoxia-responsive characteristics of PTX-NB, MTPE-TT and PTX-NB were co-loaded to self-assemble into MPNPs with the

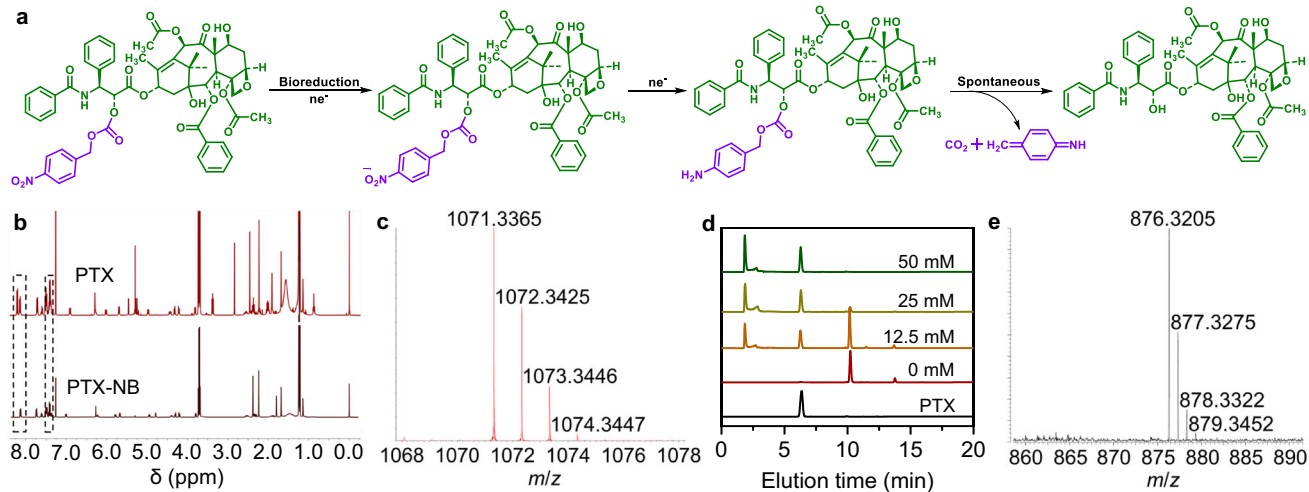

**Fig. 3 | The release of PTX from prodrug PTX-NB. a** The proposed hypoxia-triggered release mechanism of PTX-NB prodrug. After the hypoxic bioreduction, PTX-NB was converted into the unstable amine intermediate, which would undergo spontaneous self-elimination to release PTX. **b** $^1$H NMR spectra of PTX and PTX-NB in CDCl$_3$. **c** HRMS of PTX-NB. **d** HPLC measurements demonstrating the PTX release from PTX-NB after treating with different concentrations of Na$_2$S$_2$O$_4$ as indicated. Experiment was repeated three times independently with similar results. **e** HRMS analysis of PTX-NB treated with Na$_2$S$_2$O$_4$ indicating the release of PTX. Source data are provided as a Source Data file.

aid of Pluronic F127. For control, the NPs based on PTX-NB prodrug alone were also prepared by assembly of PTX-NB with Pluronic F127, named as PNPs. To endow the NPs with tumor-targeting ability, the surface of MPNPs was further coated with M1 macrophage membrane to afford the biomimetic M1-MPNPs. M1 macrophage membrane with the associated membrane proteins were expected to serve as a concealing cloak against RES clearance and as a tumor-homing navigator to enhance the NPs accumulation in inflammatory tumor tissue[61,62]. The preparation of the membrane-coated MPNPs is illustrated in Fig. 1. First, RAW264.7 cells were stimulated for 24 h with lipopolysaccharide (LPS) (100 ng mL$^{-1}$) and type II interferon (IFN-γ) (50 ng mL$^{-1}$) to induce them into M1 phenotype. As presented in Fig. 4a, the change of cell morphology from round to fusiform ones indicated the M1 polarization of macrophages. The flow cytometry analysis also indicated that the RAW264.7 cells stimulated by LPS and IFN-γ showed very high expression level of CD86 (99.2%) (Fig. 4b), a specific surface marker of M1 macrophages, suggesting the successful transformation to M1 macrophage cells[63]. Then the membrane-decorated NPs were fabricated by repeated extrusion of MPNPs with the freshly extracted M1 macrophage membrane. For control, the M1 macrophage membrane-derived empty nanovesicles without NPs cores (M1NPs) were prepared by membrane extrusion. To confirm whether the M1 macrophage membranes contained residual LPS and/or IFN-γ that were used to generate M1 macrophages, ELISA and flow cytometry analyses were conducted. Compared with the control M0 macrophage membrane, the M1 macrophage membrane did not show any increased levels of LPS and IFN-γ (Supplementary Fig. 27). Dynamic light scattering (DLS) and transmission electron microscopy (TEM) measurements were used to investigate the size and morphology of different NPs. As shown in Fig. 4c–e and Supplementary Fig. 28, the average diameters of PNPs, MNPs, M1NPs, MPNPs, and M1-MPNPs were measured to be 83, 106, 121, 111, and 126 nm, respectively. All NPs possessed negative zeta potential below −10 eV (Fig. 4f). TEM images suggested that they were uniform sphere structures, and a thin layer of cell membrane was observed on the surface of M1-MPNPs (Fig. 4c,d). Additionally, the expression of typical M1 macrophage markers, CD86 and iNOS, on both M1 macrophage membrane and M1-MPNPs were confirmed by Western blotting analysis (Fig. 4g, Supplementary Fig. 29), which revealed the successful encapsulation of M1 macrophage membrane on the surface of M1-MPNPs. The drug loading content of PTX-NB in

M1-MPNPs was calculated to be about 8.3% by HPLC analysis. The NPs also possessed good colloidal stability and nearly no diameter changes were observed after storage in PBS and serum for four days (Supplementary Fig. 30). In addition, considering the weak acidic environment of tumor (pH around 6.5–7.0), we also investigated the stability of M1-MPNPs at pH 6.5, and they also showed good stability in this weak acidic environment, which was consistent with previous studies[64]. In the future, we will conduct in-depth investigations to study the stability and integrity of membrane-coated NPs in more complicated physiological conditions. As shown in Fig. 4h, the excitation-emission mapping of M1-MPNPs indicated that it could be excited by a wide spectral region of light from 550 to 700 nm, and exhibited bright NIR fluorescence emission in the range of 750–1000 nm.

**In vitro cellular investigation**

Considering the excellent light-induced ROS production and chemotherapy properties of M1-MPNPs under hypoxic conditions, we then examined its anti-tumor ability in vitro. The uptake of MPNPs and M1-MPNPs by different cells was first evaluated by a confocal laser scanning microscope (CLSM). As displayed in Fig. 5a, the CLSM images of 4T1 cancer cells cultured with MPNPs or M1-MPNPs (5 μg mL$^{-1}$) at 37 °C for 4 h showed that a larger number of M1-MPNPs could be internalized by 4T1 cancer cells as compared to MPNPs without M1 macrophage membrane modification. Quantitative calculation indicated that the red fluorescence intensity from M1-MPNPs-treated cells was nearly two times higher than that of MPNPs-treated cells, which suggested good tumor cell targeting ability of the M1 membrane camouflage. However, when these two kinds of NPs were incubated with RAW264.7 macrophage cells at 37 °C for 4 h, CLSM images revealed the opposite results as weaker fluorescence signal was observed in the M1-MPNPs-treated cells (Fig. 5b, c). The uptake of M1-MPNPs by healthy epithelial cells was also evaluated. As presented in Supplementary Fig. 31, the human breast epithelial cells (MCF-10A) and renal proximal tubule epithelial cells (HK-2) exhibited obviously weaker fluorescence signals than 4T1 cancer cells after incubating with M1-MPNPs, suggesting that M1-MPNPs had better targeting ability toward cancer cells over healthy epithelial cells. We further tested the membrane proteins to gain in-depth understanding about this tumor-targeting phenomenon. According to previous researches, the inherited proteins from macrophage membrane, such as CD47, could

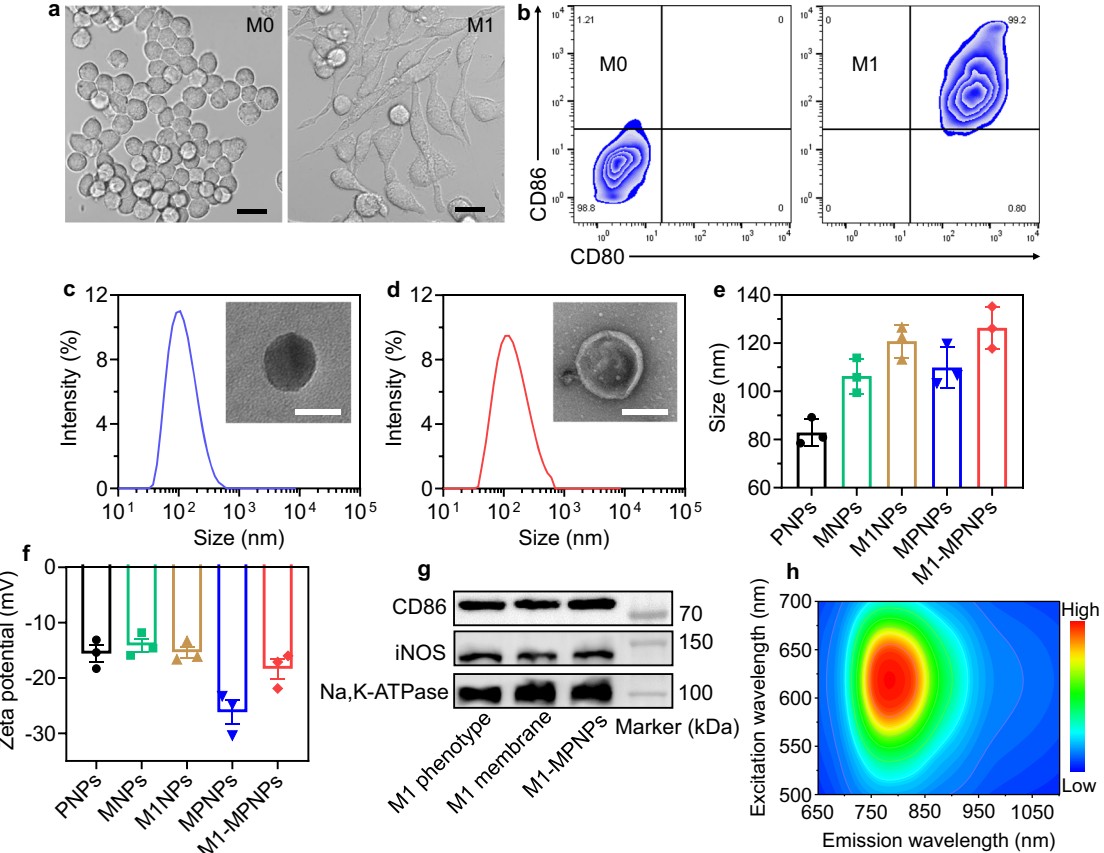

**Fig. 4 | Fabrication and characterizations of different NPs. a** Microscopic images of RAW264.7 cells before and after treating with LPS and IFN-γ. Scale bars: 20 μm. **b** Representative flow cytometry plots of M1 phenotype (CD80⁺ CD86⁺) in RAW264.7 cells before and after treating with LPS and IFN-γ. Representative DLS results and (inset) TEM images of **c** MPNPs and **d** M1-MPNPs. Scale bars: 100 nm.

**e, f** Average sizes and zeta potentials of various NPs. Data are presented as mean ± SD ($n = 3$ independent experiments). **g** Representative western blots of CD86 and iNOS expression in different formulations. **h** The excitation-emission mapping of M1-MPNPs. For **b–d, g, h**, experiment was repeated three times independently with similar results. Source data are provided as a Source Data file.

prevent undesirable macrophage-mediated phagocytosis by binding SIRPα expressed on macrophages[65]. On the other hand, α4 and β1 integrins on macrophage cell membrane could actively bind to the vascular cell adhesion molecule-1 (VCAM-1) on cancer cells, endowing them with tumor-targeting ability[66]. As shown in the western blotting analysis, we also observed the expression of these related proteins, including CD47, α4 and β1 integrins on both M1 macrophage membrane and M1-MPNPs (Supplementary Fig. 32). In further, we will conduct more studies to investigate the mechanism behind the differences in cellular uptake.

We next used DCF-DA to evaluate the intracellular ROS generation capability of different NPs. Both MNPs and M1-MPNPs (5 μg mL⁻¹) were able to generate a high concentration of ROS under white light irradiation (10 mW cm⁻², 3 min) as the treated 4T1 cells showed bright green fluorescence signal from the product of DCF-DA (Fig. 5d), confirming the excellent PDT property of MTPE-TT. Then, we investigated the activation of PTX-NB prodrug in tumor cells using HPLC analysis. The results indicated that PDT process could accelerate the conversion of PTX-NB to free PTX in cells (Supplementary Fig. 33). 3-(4,5-Dimethylthiazol-2-yl)2,5-diphenyl tetrazolium bromide (MTT) assays were then performed to examine the cytotoxicity of various NPs under different conditions. The 4T1 cells were incubated with PBS, MPNPs, or M1-MPNPs at different concentrations, followed by no treatment or light exposure. The light irradiation alone showed negligible cytotoxicity, while the cellular viabilities in other groups all exhibited concentration-dependent decrease manners. In particular, the viability of the cells treated with M1-MPNPs plus light was demonstrably lower

than that of M1-MPNPs and "MPNPs + L" (Fig. 5e, Supplementary Fig. 34). The cell death was further evaluated using annexin-V and Sytox Green co-staining method with flow cytometry analysis. Annexin-V staining is commonly used for detecting phosphatidylserine exposure on apoptotic cells, and Sytox dye can permeate dead cells to stain them with intense fluorescence by binding to cellular nucleic acids. As demonstrated in Supplementary Figs. 35 and 36, the extent of Annexin V⁺ and Sytox Green⁺ cells greatly increased with the treatment of "M1-MPNPs + L", suggesting high level of cell apoptosis and death. The results were consistent with the MTT assay, revealing potent tumor cell-killing ability of M1-MPNPs under light irradiation. Furthermore, we studied the types of regulated cell death using different kinds of cell death inhibitors. For example, Liproxstatin-1 (Lip-1), z-VAD-fmk, and Necrostatin 1s (Nec1s) were used as the inhibitors of ferroptosis, apoptosis and necroptosis, respectively[67]. As shown in Supplementary Fig. 37, the cell death caused by "M1-MPNPs + L" was only slightly inhibited by Lip-1 treatment. However, z-VAD-fmk or Nec1s significantly inhibited the cell death, suggesting that apoptosis and necroptosis were the two main forms of cell death involved in the treatment of this work. In addition, to demonstrate that the PTX-NB prodrug-based nanoplatforms had reduced side effect on normal cells, M1-MPNPs were incubated with MCF-10A epithelial cells. For control, the free PTX-based NPs (M1-MFNPs) were prepared by using free PTX to replace PTX-NB prodrug. The MTT assay indicated that M1-MPNPs showed obviously reduced cytotoxicity toward MCF-10A cells than M1-MFNPs (Supplementary Fig. 38). Taken together, these results revealed that M1-MPNPs that combined PDT, chemotherapy, and tumor cell

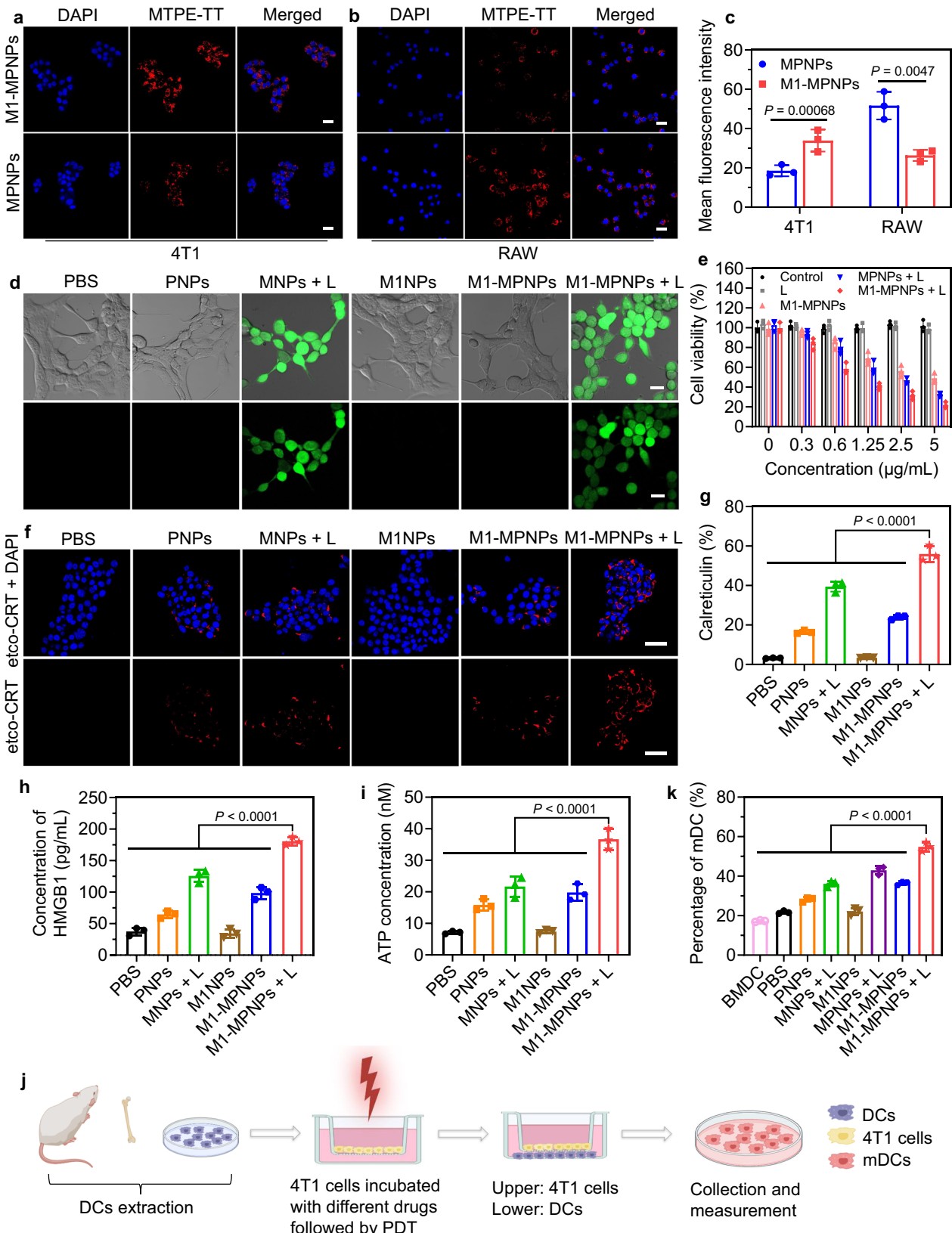

targeting were a kind of potent agents for significantly killing cancer cells under light trigger.

In addition to directly killing tumor cells, we next investigated whether M1-MPNPs could be used as a potent ICD inducer to enhance tumor immunogenicity. The immunofluorescence staining showed that both PNPs and "MNPs + L" could increase the expression

of ecto-CRT on cancer cells, indicating that the single chemodrug or PDT was able to trigger ICD. For the 4T1 cancer cells treated with M1-MPNPs plus white light irradiation (10 mW cm$^{-2}$), the ecto-CRT expression was about 2.77-fold and 1.31-fold higher than that of PNPs and "MNPs + L" groups (Fig. 5f, Supplementary Fig. 39), respectively. The calreticulin exposure was also analyzed by flow cytometry, and

**Fig. 5 | In vitro cellular investigation.** Representative CLSM images of **a** 4T1 cells and **b** RAW cells upon incubation with MPNPs or M1-MPNPs. Scale bars: 20 μm. **c** Quantitative data showing the mean fluorescence intensity (MFI) based on Fig. 5a, b. Data are presented as mean ± SD (*n* = 3 independent experiments). Statistical significance was determined using two-tailed Student's *t* test. **d** Representative CLSM images showing the ROS generation in 4T1 cancer cells with different treatments as indicated (DCF-DA was nonfluorescent but switched to fluorescent DCF with green signal when oxidized by ROS). Scale bars: 20 μm. Experiment was repeated three times independently with similar results. **e** Cell viabilities of 4T1 cancer cells treated with various NPs without or with light irradiation in hypoxic environment. Data are presented as mean ± SD (*n* = 3 independent experiments). **f** Representative CLSM images showing the expression of ecto-CRT (red fluorescence) on 4T1 cancer cells surface after different treatments. The cell nuclei were stained with 4',6-diamidino-2-phenylindole (DAPI; blue fluorescence). Scale bars: 50 μm. Experiment was repeated three times independently with similar results. **g** Quantitative analysis of the calreticulin exposure after various treatments using flow cytometry. The histograms represented the calreticulin detected in non-permeabilized (Sytox Green⁻) cells. Data are presented as mean ± SD (*n* = 3 independent experiments). Statistical significance was determined using one-way ANOVA. **h** Analysis of HMGB1 concentration in the supernatant of 4T1 tumor cells after different treatments by ELISA assay. Data are presented as mean ± SD (*n* = 3 independent experiments). Statistical significance was determined using one-way ANOVA. **i** Quantitative analysis of the ATP concentration in cell supernatants of 4T1 cancer cells received various treatments as indicated. Data are presented as mean ± SD (*n* = 3 independent experiments). Statistical significance was determined using one-way ANOVA. **j** Schematic illustration of the measurement of BMDCs maturation. The illustration was created with BioRender.com. **k** Quantitative analysis of the percentage of BMDCs maturation upon incubation with tumor cells subjected to various treatments. Data are presented as mean ± SD (*n* = 3 independent experiments). Statistical significance was determined using one-way ANOVA. Source data are provided as a Source Data file.

Sytox Green was used as the permeabilization marker to exclude internal staining. The results indicated that "M1-MPNPs + L" treatment resulted in the highest level of calreticulin exposure (Fig. 5g, Supplementary Fig. 40). Furthermore, we also demonstrated that the supernatant of the 4T1 cells treated with "M1-MPNPs + L" contained a larger number of other DAMPs, including HMGB1 and ATP (Fig. 5h, i). As the AIEgen-based PDT effect could not only induce ICD but also accelerate the hypoxia-responsive release of PTX drug by consuming the local oxygen, thus the combination of PDT and hypoxia-responsive prodrug was capable of achieving self-accelerated ICD induction with synergistic effect of "1 + 1 > 2". These results suggested that M1-MPNPs could strongly induce ICD of tumor cells, which was essential to enhance the tumor antigen presentation and T lymphocyte activation.

We further analyzed the maturation of bone-marrow-derived dendritic cells (BMDCs) to validate the ICD effect induced by M1-MPNPs. As schematically illustrated in Fig. 5j, BMDCs were isolated from the tibia and femur of BALB/c female mice (7 weeks old) according to previous reseach[68]. Then 4T1 cancer cells were seeded onto the upper chamber of transwells, and treated with "PBS", "PNPs", "MNPs + L", "M1-PNPs", "MPNPs + L", "M1-MPNPs", and "M1-MPNPs + L", respectively. Afterward, BMDCs were incubated in the lower transwell compartment. Finally, BMDCs were collected and stained by anti-CD11c, anti-CD80, and anti-CD86, and detected using a flow cytometer. Impressively, co-incubation of BMDCs with 4T1 cancer cells pretreated by "M1-MPNPs + L" caused obviously increased expression of CD80 and CD86 (Fig. 5k, Supplementary Figs. 41 and 42), suggesting strong DC maturation.

## In vivo fluorescence and PA imaging of tumor-bearing mice

The in vivo tumor imaging ability and biological distribution of M1-MPNPs were then examined. After intravenous injection of MPNPs or M1-MPNPs into the tail vein of 4T1 tumor-bearing mice, the mice were imaged under in vivo imaging system (IVIS). Significant NIR fluorescence signal from MTPE-TT was observed in the tumor site, and the fluorescence intensity reached maximum at about 24 h post-injection (Fig. 6a), which suggested the optimal time point for tumor photo-imaging and treatment. Interestingly, the fluorescence signal from the mice treated with M1-MPNPs was about 1.6-fold higher than that of MPNPs (Fig. 6b), suggesting the better tumor accumulation ability mediated by M1 macrophage membrane coating. In addition, even at 48 h after administration, a relatively strong NIR fluorescence signal could still be observed at the tumor site, indicating the long-term in vivo tumor imaging capability of the NPs. Fluorescence imaging possesses high sensitivity, but the penetration depth and spatial resolution are limited. By contrast, PA imaging can provide large penetration depth beyond the optical diffusion limit meanwhile maintaining high spatial resolution. As M1-MPNPs showed strong absorption and PA signal in NIR spectral region, we next conducted in vivo PA imaging to gain more detailed information about the tumor. As displayed in Supplementary Fig. 43, the PA amplitude of tumor exhibited a similar time-dependent profile as the fluorescence imaging, which also became highest at 24 h after intravenous administration. The PA image under 680 nm excitation helped to delineate the location and shape of the tumor region in a high-contrast manner (Fig. 6c). Accordingly, the in vivo fluorescence and PA imaging provided comprehensive information about the tumor site, which would be useful to guide the treatment process. The biodistribution of M1-MPNPs was also studied. After the NPs were intravenously injected into tumor-bearing mice for 24 h, the main organs and tumors were resected and imaged under IVIS. The tumors of M1-MPNPs-treated mice exhibited much higher NIR fluorescence signal than that of MPNPs, while the liver and spleen possessed decreased NPs accumulation (Fig. 6d, e), probably due to that the M1 macrophage cell membrane could help NPs to escape RES. This was consistent with previous cellular experiments that M1-MPNPs showed less uptake in RAW264.7 cells. To examine the location of NPs within tumor tissue, tumors from the mice receiving different NPs treatments were sectioned and observed under fluorescence microscope. The tumor sections were also stained with anti-CD31 antibodies to visualize the blood vessels. It was interesting to note that the NIR fluorescence signal of the M1-MPNPs-treated tumor was stronger than that of MPNPs group, especially in the deep tumor site (≈2.5 mm) (Fig. 6f, g). The deep tumor site of mice injected with M1-MPNPs exhibited about 7-fold higher NIR fluorescence signal when compared with that of the MPNPs-treated ones. The excellent tumor penetration ability of M1-MPNPs was probably due to the enhanced tumor cell targeting mediated by M1 macrophage membrane camouflage.

## Photo-triggered self-accelerated M1-MPNPs for synergistic immunotherapy of tumor

First, the tumor prophylactic vaccination experiment was conducted to test whether the "M1-MPNPs + L"-induced ICD of tumor cells had immunogenicity in vivo. As shown in Supplementary Fig. 44, the 4T1 tumor cells pretreated with "M1-MPNPs + L" were subcutaneously injected into the left flank of mice on day 0 to establish anti-tumor immunity. The mice injected with PBS were subjected as control. On day 7, the mice were challenged with tumor cells by subcutaneous injection of live 4T1 tumor cells into the right flank of mice. The tumor growth on the right site was monitored every two days. As presented in Supplementary Fig. 44, the prophylactic vaccination with 4T1 cells killed by "M1-MPNPs + L" could protect the mice against the subsequent tumor cell rechallenge, and the tumor growth was greatly inhibited compared with PBS group. After confirming the excellent ICD induction and tumor imaging ability of M1-MPNPs, we next investigated their immunotherapeutic properties in 4T1 tumor-bearing BALB/c mice. As shown in Fig. 7a, seven days after the tumor

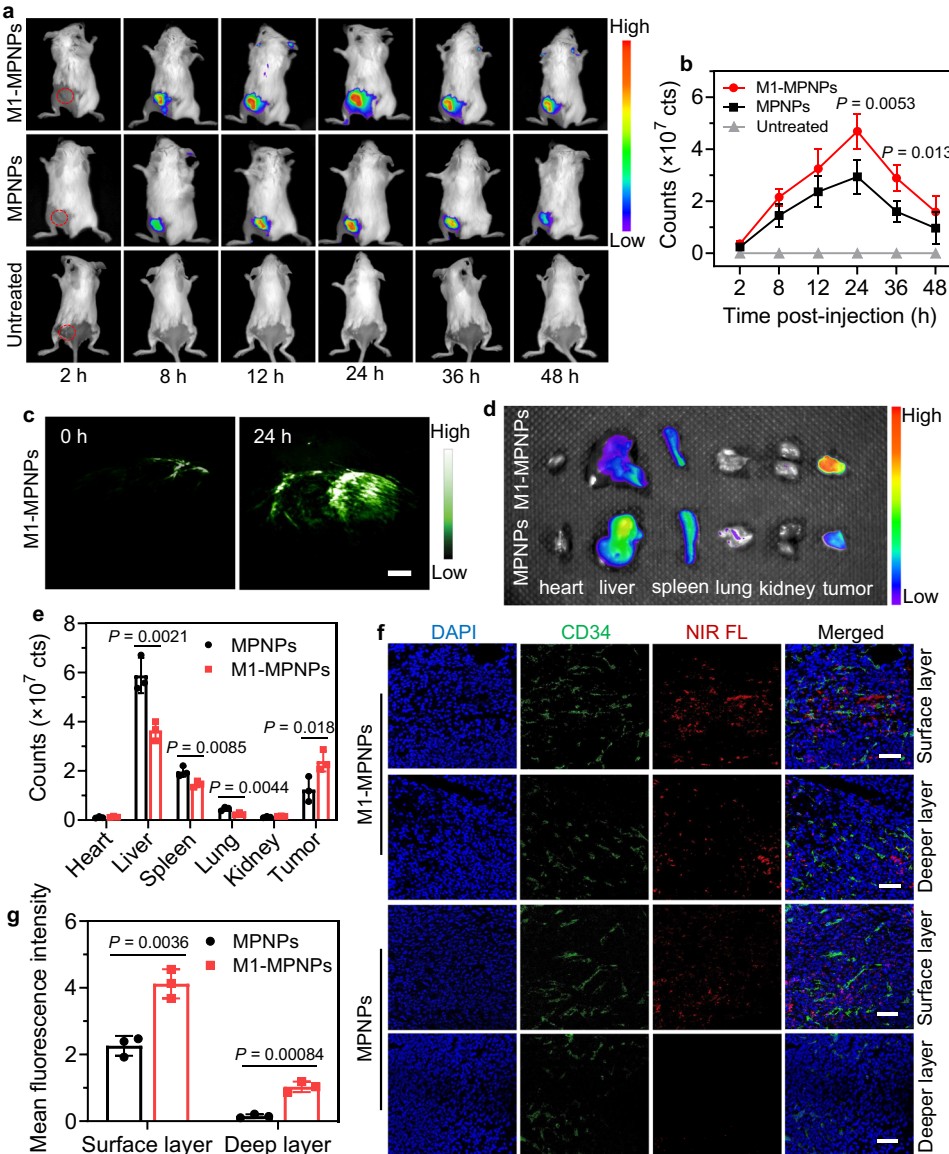

**Fig. 6 | In vivo fluorescence and PA imaging of tumor-bearing mice.**
**a**, **b** Representative fluorescence images of 4T1 tumor-bearing mice and corresponding fluorescence intensities in the tumor regions at various time points post i.v. injection of MPNPs or M1-MPNPs, the tumor-bearing mice with no treatment were used as control. Data are presented as mean ± SD ($n$ = 3 mice). Statistical significance was determined in comparison with "MPNPs" group via one-way ANOVA. **c** Representative in vivo PA images of tumor site at different time points post i.v. injection of M1-MPNPs. Scale bars: 2 mm. Experiment was repeated three times independently with similar results. **d**, **e** Representative ex vivo fluorescence imaging and corresponding fluorescence intensities of major organs and tumors isolated from mice at 24 h post i.v. injection of MPNPs or M1-MPNPs, the organs from the tumor-bearing mice without any treatment were used as control. Data are presented as mean ± SD ($n$ = 3 mice). Statistical significance was determined using one-way ANOVA. **f** Representative fluorescent images of tumor sections from the mice injected with MPNPs or M1-MPNPs. Blue: DAPI for staining cell nucleus; green: CD34 for staining tumor neovasculature; and red: NPs. Scale bars: 50 μm. **g** Quantitative data showing the mean fluorescence intensity (MFI) based on Fig. 6f. Data are presented as mean ± SD ($n$ = 3 mice). Statistical significance was determined using two-tailed Student's $t$ test. Source data are provided as a Source Data file.

inoculation (tumor volume ≈ 80 mm³), the tumor-bearing mice were randomly divided into seven groups ($n$ = 5 mice per group), including PBS, PNPs, MNPs + L, M1NPs, MPNPs + L, M1-MPNPs, and M1-MPNPs + L. At 24 h after intravenous injection of different NPs (200 μL, 500 μg mL⁻¹), the tumor sites of the groups with "L" were irradiated with white light (0.3 W cm⁻²) for 10 min, which was conducted twice on day 0 and day 3. The tumor volumes of the mice receiving various treatments were monitored every other day. As shown in Fig. 7b,c, the tumor growth of "M1-MPNPs + L" group was greatly inhibited, and the mean tumor volume on day 14 was only 7.6 mm³, which was much smaller than that of "MPNPs + L" group (242.7 mm³) and PBS group (1545.6 mm³). The ultimate average tumor weight of the mice treated with "M1-MPNPs + L" was only 53 mg (Fig. 7d), which was about 22-, 15-, 5-, 20-, 9- and 12-times smaller than that of PBS (1168 mg), PNPs (798 mg), MNPs + L (266 mg), M1NPs (1053 mg), MPNPs + L (477 mg), and M1-MPNPs (629 mg) groups, respectively. In addition, there was negligible difference in the body weight of the mice receiving various treatments (Fig. 7e), suggesting the good biosafety of all treatments. The superior antitumor efficacy of M1-MPNPs plus light mainly stemmed from their potent PDT effect and the self-accelerated activation of the hypoxia-responsive PTX prodrug, as well as the M1 macrophage cell membrane-enhanced tumor homing.

The histological hematoxylin and eosin (H&E) staining and terminal deoxynucleotidyl transferase (TdT)-mediated dUTP-biotin nick

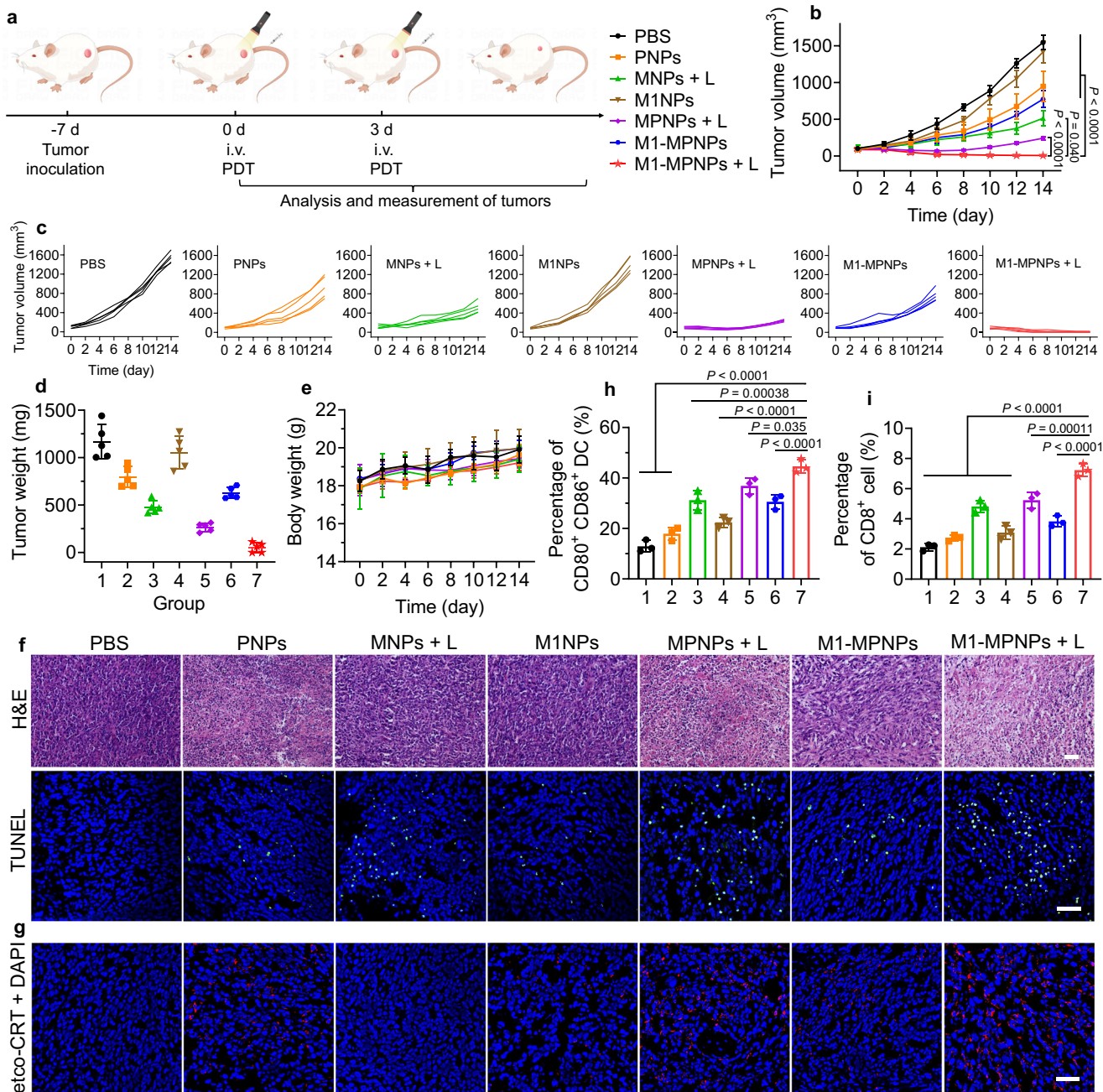

**Fig. 7 | Photo-triggered self-accelerated M1-MPNPs for synergistic immunotherapy of tumor. a** Experimental outline showing the treatment steps and procedures for evaluating the therapeutic outcomes in 4T1 tumor-bearing mice. The "i.v." represents "intravenous injection". The illustration was created with BioRender.com. Plots of **b** tumor volume, **c** individual tumor volume, **d** tumor weight, and **e** body weight of the 4T1 tumor-bearing mice with various treatments. Data are presented as mean ± SD ($n = 5$ mice). Statistical significance was determined using one-way ANOVA. **f** Representative H&E staining and TUNEL staining of tumor sections harvested from the mice receiving different treatments on day 14 ($n = 3$ mice). Scale bars: 50 μm. Experiment was repeated three times independently with similar results. **g** Representative CLSM images showing ecto-CRT expression (red pseudocolor) on the tumor sections from the mice receiving various treatments. Scale bars: 50 μm. Experiment was repeated three times independently with similar results. **h** Quantitative data of the population of DC maturation (CD11c+CD80+CD86+) in lymph nodes after various treatments. Data are presented as mean ± SD ($n = 3$ mice). Statistical significance was determined using one-way ANOVA. **i** Quantitative data of the percentages of CD8+ T cells in tumor. Data are presented as mean ± SD ($n = 3$ mice). Statistical significance was determined using one-way ANOVA. Source data are provided as a Source Data file.

end labeling (TUNEL) staining were performed on tumor tissue sections to evaluate the apoptosis and necrosis after different treatments. The results suggested that larger necrotic areas and more serious tumor cell apoptosis were observed in the "M1-MPNPs + L" group (Fig. 7f). In addition, the etco-CRT expression on tumor tissues of each group was evaluated by immunofluorescence staining. As shown in Fig. 7g, the tumor tissue in "M1-MPNPs + L" group again exhibited the highest levels of CRT expression, indicating that "M1-MPNPs + L"

treatment could efficiently induce ICD in vivo. The ICD of tumor cells may greatly promote the maturation of DCs by improving tumor antigen presentation, and DCs' maturation is a key step to generate effective antigen-specific immunity. Thus, we evaluated the presence of immune cells in tumors and lymph nodes adjacent to tumors. First, the levels of DCs maturation in lymph were determined by flow cytometer analysis. The percentage of mature DCs (CD80+CD86+ gated on CD11c+) in the "M1-MPNPs + L" group was about 43%, which was

significantly higher than that of PBS (13.2%), PNPs (19.7%), MNPs + L (30.5%), M1NPs (23.1%), MPNPs + L (37.5%), and M1-MPNPs (28.1%) groups (Fig. 7h, Supplementary Figs. 45 and 46). As a result of the efficient maturation of DCs, CD8$^+$ cytotoxic T lymphocytes could also be activated, which were the most powerful effectors in the anticancer immune response. As presented in Fig. 7i and Supplementary Fig. 47, the "M1-MPNPs + L" treatment resulted in 1.38−3.41-folds higher infiltration of CD8$^+$ T cells in tumors as compared with other groups, suggesting that it could lead to more effective antitumor immune response. Taken together, these results indicated that M1-MPNPs were able to induce ICD by combining the high ROS generation ability and self-accelerated hypoxia-triggered PTX release under light trigger, which thus elicited superior immune response and excellent antitumor outcomes.

## M1-MPNPs for suppressing primary tumor and distant tumor recurrence

We next investigated whether such strong anti-tumor immune response induced by M1-MPNPs plus light would exert abscopal inhibition effect against the untreated distant tumors. For this, a bilateral 4T1 tumor model was established as illustrated in Fig. 8a. Specifically, the left side of each BALB/c mouse was subcutaneously injected with $1 \times 10^6$ of 4T1 cells to establish the primary tumor. Six days later, the right side of each mouse was inoculated with a second tumor as the distant tumor to simulate the metastatic tumor. Then, seven days after primary tumor inoculation, the bilateral 4T1 tumor-bearing mice were also randomly divided into seven groups (n = 5 mice per group) as follows: PBS, PNPs, MNPs + L, M1NPs, MPNPs + L, M1-MPNPs, and M1-MPNPs + L. On day 0 and day 3, different NPs (200 μL, 500 μg mL$^{-1}$) were injected into the mice through tail vein, and only primary tumors in the groups with "L" were exposed to white light treatment (0.3 W cm$^{-2}$, 10 min) at 24 h post NPs administration, while the distant tumors were left untreated. As depicted in Fig. 8b−d and Supplementary Fig. 48, MPNPs with light irradiation and M1-MPNPs without light irradiation only exhibited moderate inhibition on the tumor growth of both sides, demonstrating that PDT or chemodrug alone could not efficiently suppress tumors. Nevertheless, both primary and distant tumors were significantly inhibited by the "M1-MPNPs + L" treatment. Impressively, the average volumes of distant tumors in "MPNPs + L", "M1-MPNPs" and "PBS" groups were in sequence 2.53, 7.61, and 12.18 times larger than that of "M1-MPNPs + L" group on day 14 (Fig. 8c). All these treatments showed no obvious influence on the body weight of mice (Fig. 8e). These results confirmed that the combination of potent PDT effect and PDT-accelerated chemotherapy was beneficial to strong suppression on both primary and distant tumors.

To elucidate the mechanisms under the superior antitumor efficacy of M1-MPNPs, immunofluorescent staining and flow cytometry analysis were conducted to measure the immune responses on day 14. Immunofluorescence staining results showed that "MPNPs + L", M1-MPNPs and "M1-MPNPs + L" were all able to evoke ICD of tumor in vivo, among which "M1-MPNPs + L" group showed the strongest ICD induction effect. This was demonstrated by the extensive exposure of CRT protein on the surface of tumor cells (Supplementary Figs. 49 and 50). Then, we investigated the maturation of DCs in the spleen of mice and the infiltration of relevant immune cells in bilateral tumors. We consistently found that M1-MPNPs could greatly increase the proportions of mature DCs (CD80$^+$CD86$^+$ cells) in lymph nodes under light irradiation (Fig. 8f, g). The percentages of CD8$^+$ T cells in both primary and distant tumors of "M1-MPNPs + L" group were also obviously increased (Fig. 8h−j, Supplementary Figs. 51 and 52). As depicted in Fig. 8j, the percentage of CD8$^+$ T cells in the distant tumors of "M1-MPNPs + L" group was 1.45 times and 2.26 times higher than that of "MPNPs + L" group and "M1-MPNPs" group, respectively, suggesting the strong anti-tumor

immune effect induced by M1-MPNPs. In addition to activating immune responses, inducing immunological memory is very important for exerting a long-term protection against tumor recurrence[69,70]. Therefore, we evaluated whether immunological memory was established in the bilateral tumors-bearing mice treated with "M1-MPNPs + L" by measuring the expression of effector memory T cells (Tem, CD44$^+$CD62L$^-$ cells) in the spleens of mice. Encouragingly, the flow cytometry analyses indicated that the percentage of CD8$^+$ Tem cells (CD44$^+$ and CD62L$^-$) in the spleen of "M1-MPNPs + L" group (29.3%) was significantly higher than that of PBS-treated control group (15.9%) (Fig. 8k, l, Supplementary Fig. 53). H&E and TUNEL staining of distant tumors also suggested larger scales of necrotic areas and tumor cell apoptosis in the "M1-MPNPs + L" group. These results manifested that M1-MPNPs was a kind of potent ICD inducer that could elicit strong anti-tumor immunity.

Finally, the in vivo safety profile of M1-MPNPs was evaluated. Healthy mice were randomly divided into two groups (n = 3 mice), and PBS or M1-MPNPs were injected intravenously on day 0 and day 3, respectively. The major organs of mice, including heart, liver, spleen, lungs, and kidneys, were harvested on day 10 for pathological analysis. As shown in Supplementary Fig. 54, the H&E-stained slices of the major organs from both PBS- and M1-MPNPs-treated mice showed negligible damage and impairment. The weight of the spleens from mice with the treatment of PBS and M1-MPNPs was also similar (Supplementary Fig. 55). Moreover, the blood samples were also collected on day 10 for biochemical analyses. As depicted in Supplementary Figs. 56 and 57, the blood routine examination parameters and hepatic/renal function indicators indicated no noticeable difference between the PBS- and M1-MPNPs-treated mice. Taken together, these results implied that M1-MPNPs possessed good biocompatibility.

## Discussion

In this work, we have developed a kind of organic phototheranostic platform by incorporating a NIR AIEgen and hypoxia-responsive prodrug, which enables fluorescence and PA dual-mode imaging of tumor, as well as combination immunotherapy mediated by the potent PDT effect and self-accelerated chemotherapy. It is validated that methoxy substitution of TPE represents an efficient strategy to increase the AIE characteristic and molar absorption coefficient, simultaneously boosting the fluorescence and PA brightness. Noteworthy, both type I and type II PDT processes are amplified in MTPE-TT via the more efficient ISC transition. The high-performance AIEgen and PTX-based hypoxia-responsive prodrug are formulated into organic NPs, which could be further camouflaged with M1 macrophage cell membrane to increase the tumor-targeting ability. The complementary merits of fluorescence and PA imaging allow for sensitive delineation of the tumor site, providing accurate guidance for phototherapy. The light-generated ROS is capable of inducing ICD of tumor cells, during which the oxygen consumption leads to severe hypoxia to trigger the release of PTX to enhance the antitumor immune responses. Benefitting from the excellent PDT property and self-accelerated chemotherapeutic prodrug, this nanoplatform is capable of eliciting strong antitumor immunity to suppress both the primary tumor and distant tumor. This work puts forward an idea that methoxy substitution represents an effective method to increase the fluorescence brightness by intensified AIE effect, boost the PA intensity through amplified light absorption ability, and promote both type I and type II PDT via enhanced ISC process. The self-accelerated strategy by integrating PDT agent and hypoxia-responsive prodrug represents an efficient way to develop robust immunotherapy agents for boosting ICD-based antitumor treatment. Therefore, this work will provide insights into the development of high-performance organic theranostic agents and multifunctional imaging-guided cancer immunotherapy.

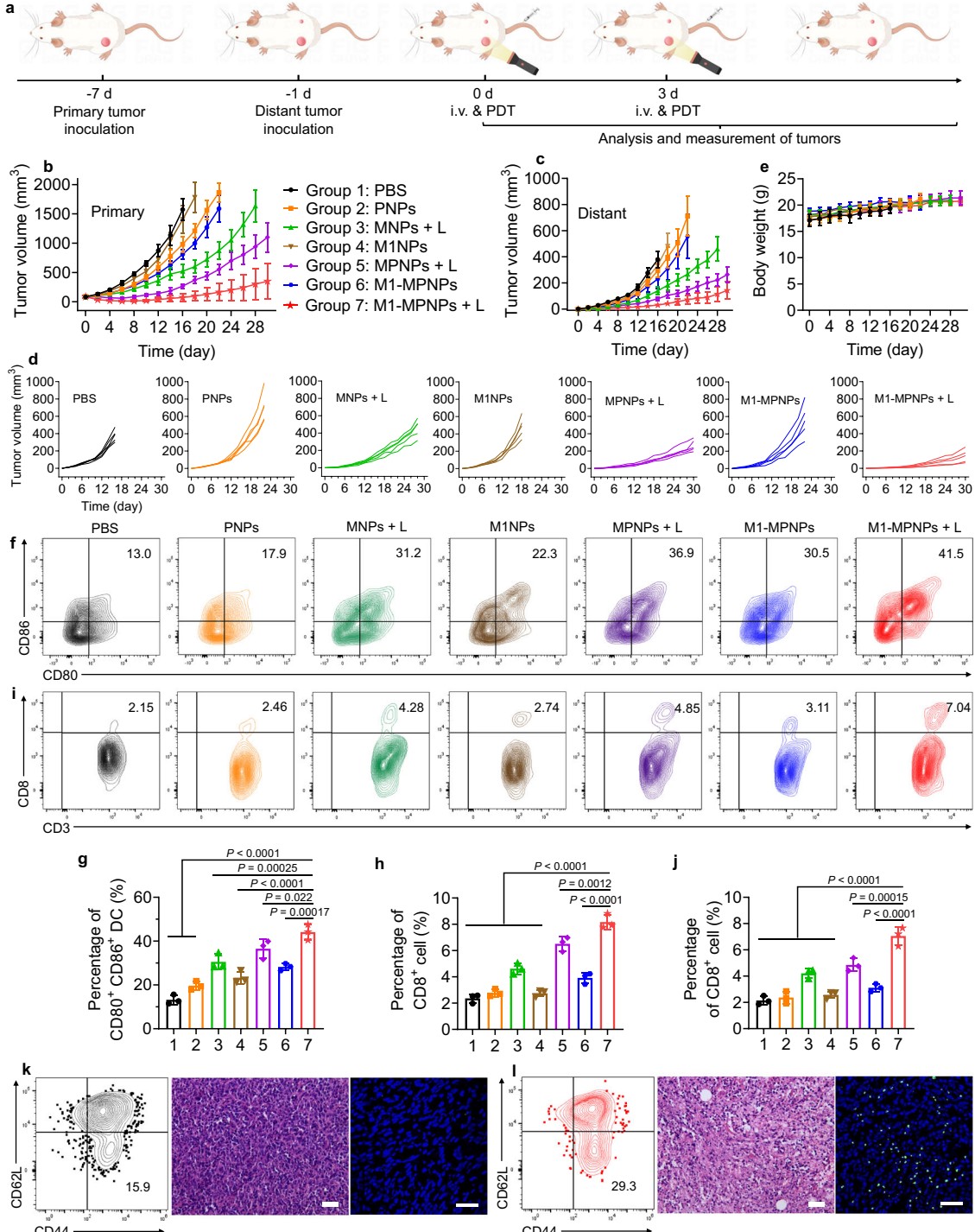

**Fig. 8 | M1-MPNPs for suppressing primary tumor and distant tumor recurrence. a** Schematic illustration of the procedure for evaluation of the antitumor effect and immune responses induced by various treatments in bilateral 4T1 tumor-bearing mice model. The "i.v." represents "intravenous injection". The illustration was created with BioRender.com. Plots of **b** primary tumor volume, **c** distant tumor volume, **d** individual tumor growth curve of distant tumor, and **e** body weight of the bilateral 4T1 tumor-bearing mice post various treatments. Data are presented as mean ± SD (*n* = 5 mice). **f, g** Representative flow cytometry analysis and quantitative data of the population of DC maturation (CD11c⁺CD80⁺CD86⁺) in lymph nodes after various treatments on day 14. Data are presented as mean ± SD (*n* = 3 mice). Statistical significance was determined using one-way ANOVA. **h** Quantitative data of the percentages of CD8⁺ T cells in primary tumors collected from bilateral 4T1

tumor-bearing mice after various treatments on day 14. Data are presented as mean ± SD (*n* = 3 mice). Statistical significance was determined using one-way ANOVA. **i, j** Representative flow cytometry plots and quantitative data of the percentages of CD8⁺ T cells in distant tumors collected from bilateral 4T1 tumor-bearing mice after various treatments on day 14. Data are presented as mean ± SD (*n* = 3 mice). Statistical significance was determined using one-way ANOVA. Representative flow cytometry data of CD8⁺ Tem cells (CD44⁺ and CD62L⁻) in spleen, H&E and TUNEL staining of distant tumors in bilateral 4T1 tumor-bearing mice from **k** "PBS" group and **l** "M1-MPNPs + L" group on day 14 (*n* = 3 mice). Scale bars: 50 μm. Experiment was repeated three times independently with similar results. Source data are provided as a Source Data file.

## Methods

### Materials

All chemicals and reagents were provided by commercial sources. Roswell Park Memorial Institute (RPMI) 1640 culture medium, Dulbecco's Modified Eagle's Medium (DMEM) culture medium, fetal bovine serum (FBS), and penicillin streptomycin were purchased from Gibco-BRL (Grand Island, NY, USA). Lipopolysaccharide (LPS) and interferon-γ (IFN-γ) were obtained from eBioscience. Sytox Green was obtained from Thermo Fisher Scientific. Cell death inhibitors were obtained from MedChemExpress. ELISA kits were obtained from Elabscience or Cusabio. Antibodies used: Western blotting: Anti-CD86 (C-terminal) Polyclonal antibody (1:1000, Proteintech, Rabbit mAb, #26903-1-AP); Recombinant Anti-iNOS antibody (1:1000, Abcam, Rabbit mAb, #ab178945); CD47 (D3O7P) Rabbit mAb (1:1000, Cell Signaling Technology, #63000); Na,K-ATPase Antibody (1:1000, Cell Signaling Technology, #3010); Integrin alpha 4/CD49D (1:1000, Abcam, Rabbit mAb, #ab81280); Integrin beta 1 (1:1000, Abcam, Rabbit mAb, #ab52971). Immunofluorescence analysis: Recombinant Anti-CD34 antibody (1:200, Abcam, Rabbit mAb, #ab81289); Recombinant Anti-Calreticulin antibody (1:200, Abcam, Rabbit mAb, #ab92516); Goat Anti-Rabbit IgG H&L (Alexa Fluor® 488) (1:1000, Abcam, #ab150077); Donkey Anti-Rabbit IgG H&L (Alexa Fluor® 647) (1:1000, Abcam, #ab150075). Flow cytometry: PE anti-mouse IFN-γ antibody (1:200, Biolegend, #505807); PE anti-mouse/human CD44 Antibody (1:200, Biolegend, #103008); PE/Cyanine7 anti-mouse CD62L Antibody (1:200, Biolegend, #104418); FITC anti-mouse CD11c Antibody (1:200, Biolegend, #117306); APC anti-mouse CD86 Antibody (1:200, Biolegend, #105012); PE anti-mouse CD80 Antibody (1:200, Biolegend, #104708); FITC anti-mouse CD3ε Antibody (1:200, Biolegend, #100306); Brilliant Violet 421™ anti-mouse CD4 Antibody (1:200, Biolegend, #100438); APC anti-mouse CD8a Antibody (1:200, Biolegend, #100712).

### Characterizations

$^1$H and $^{13}$C nuclear magnetic resonance (NMR) spectra were recorded on Bruker-DPX 400 spectrometer. High-resolution mass spectra (HRMS) were conducted on a Bruker AutoflexIII LRF200-CID mass spectrometer in matrix assisted laser desorption ionization time of flight (MALDI-TOF) mode. The absorption spectra were measured using Shimadzu UV-1800 spectrometer. Horiba Fluorolog-3 spectrofluorometer was used to record photoluminescence (PL) spectra. Transmission electron microscope (TEM) images were captured by a Transmission electron microscopy (TEM, Talos L120C G2, FEI, Czech). HPLC analysis was carried out on a LUMTECH HPLC (Germany) system. Dynamic light scattering (DLS) measurement was performed on a Malvern Zeta sizer Nano ZS-90. Molecular geometry optimization was calculated using the DFT method with the Gaussian 09 program package (revision D. 01) at the level of B3LYP/6-31G*, and the Cartesian coordinates were presented in Supplementary Tables 1 and 2. PA imaging was conducted on a commercial small-animal opt-acoustic tomography system (MOST, iTheraMedical, Germany). In vivo imaging system (NightOWL II LB983) was employed for fluorescence imaging of animals and organs.

### Separation of M1 macrophage membrane

First, RAW264.7 cells were stimulated for 24 h with LPS (100 ng mL$^{-1}$) and IFN-γ (50 ng mL$^{-1}$) to generate M1-polarized macrophages. After washing with PBS three times, the cells were then resuspended in PBS and incubated overnight in a hypotonic lysing buffer (4 °C). Then, a tight-fitting pestle was used to grind the cells 20 times before centrifugation (20,000 g, 25 min). The supernatant was collected and centrifuged at 100,000 g for 40 min to collect the macrophage membrane. A BCA protein assay was used to determine the protein concentration of purified macrophage membranes. To further investigate whether residual LPS and IFN-γ existed on M1 macrophage membrane, the cell membrane lysates of 1 mg M1 (or M0) macrophage membranes were analyzed using LPS ELISA kit (#CSB-E13066m, Cusabio, China) according to the manufacturer's instructions. For IFN-γ measurement, the M1 (or M0) macrophages were stained with PE anti-mouse IFN-γ antibody (1:200, Biolegend, #505807) and then analyzed by flow cytometry.

### Preparation of PNPs, MPNPs, and M1-MPNPs

In total, 1 mg of MTPE-TT, 2 mg of PTX-NB and 10 mg of Pluronic F-127 were fully dissolved in 1 mL of tetrahydrofuran (THF), and then poured into 10 mL of deionized water undergoing continuous sonication for 3 min. THF in the solution was then removed by stirring at room temperature overnight in fume hood. The resulting MPNPs were collected via centrifugation using centrifuge filters with a 100-kDa molecular weight cutoff (MWCO). For control, the NPs based on PTX-NB prodrug alone were also prepared by assembly of PTX-NB prodrug with Pluronic F127, named as PNPs. To fabricate M1-MPNPs, freshly prepared MPNPs (100 μL, 1 mg mL$^{-1}$ based on MTPE-TT) were mixed with 0.5 mg of M1 macrophage membrane under ultrasound (100 W, 50 Hz, 10 min), and then extruded with polycarbonate membrane to generate M1-MPNPs. The prepared M1-MPNPs were stored at 4 °C and used as soon as possible.

### Cells and animals

The murine macrophage cell line RAW264.7, 4T1 breast cancer cell line, the human breast epithelial cell line MCF10A and the human renal proximal tubular epithelial cell line HK-2 were purchased from Chinese Academy of Sciences Cells Bank (Shanghai, China). All cells were cultivated in a humidified atmosphere at 37 °C with 5% of CO$_2$.

Female BALB/c mice (7 weeks old) were purchased from the Laboratory Animal Center of the Academy of Military Medical Sciences (Beijing, China). The living environment of animals were maintained at a temperature of 25 °C with a 12 h light/dark cycle, with free access to standard food and water. The tumor volume was calculated referring to the equation: Volume = (Length × Width × Width)/2. The humane endpoints included tumor burden exceeding 10% of normal body weight, animal weight loss exceeds 20% of normal animal weight, ulcer at tumor growth point, and sustained self-mutilation in animals. The humane end-point was approved by Certification and Accreditation Administration of the People's Republic of China (CNCA). All procedures involving animal were conducted in accordance with the guidelines set by the Tianjin Committee of Use and Care of Laboratory Animals, and approved by the Animal Ethics Committee of Nankai University.

### Cytotoxicity study

Briefly, 4T1 cells were seeded in a 96 well plate, incubated with various concentrations of MPNPs or M1-MPNPs for 24 h at 37 °C. Then, the 4T1 cells were irradiated under white light (10 mW cm$^{-2}$) for 5 min. Alternatively, 4T1 cancer cells treated with MPNPs or M1-MPNPs were kept in the dark without exposure to light. A standard (3-(4,5dimethylthiazol-2-yl)-2,5-diphenyltetrazolium bromide) (MTT) assay was used to measure the relative cell viability at 24 h after irradiation. The cell death was also analyzed using Annexin V-mCherry/Sytox Green staining assay. The cells after different treatments were stained with Annexin V-mCherry/Sytox Green kit (#C1070M, Beyotime, China) according to the experimental procedures provided by the reagent manufacturer, and then subjected to flow cytometry analysis. For cell death inhibition experiments, ferroptosis inhibitor Liproxstatin-1 (#HY-12726, MedChemExpress, USA), apoptosis inhibitor z-VAD-fmk (#HY-16658B, MedChemExpress, USA, 10 μM) or necroptosis inhibitor Necrostatin 1 s (#HY-15760, MedChemExpress, USA, 10 μM) were added into the cell culture media.

## ROS detection in 4T1 cells

The 4T1 cancer cells were pre-seeded on the confocal chambers and cultured in serum-free DMEM medium containing M1NPs, PNPs, MNPs or M1-MPNPs (2 μg mL$^{-1}$) for 24 h at 37 °C. Afterward, the cells were washed and stained with 20 μM of 2′,7′-dichlorodihydrofluorescein diacetate (DCF-DA) in DMEM medium without serum. Following this, the cells were irradiated with white light (10 mW cm$^{-2}$) for 3 min. The cells cultured with PBS were used as a control. For CLSM imaging, the excitation of dichlorofluorescein (DCF; yielded by DCF-DA reacting with ROS) detection was 488 nm using emission at 530 ± 20 nm.

## Characterization of immunologic cell death

The 4T1 cancer cells were cultivated in confocal chambers, followed by the incubation of 2 μg mL$^{-1}$ of M1NPs, PNPs, MNPs or M1-MPNPs for 24 h at 37 °C. Followed by washing with 1 × PBS, the cells were either kept in dark or exposed to white light (10 mW cm$^{-2}$) for 3 min. After 12 h of irradiation, the cells were fixed with 4% paraformaldehyde in ice for 20 min, and then incubated with anti-calreticulin antibody (1:200, Abcam, Rabbit mAb, #ab92516) for 2 h at room temperature.

Then, the cells were further incubated with Donkey Anti-Rabbit IgG H&L (Alexa Fluor® 647) (1:1000, Abcam, #ab150075) for another 90 min at room temperature, and cells nuclei were stained by DAPI. The cells were imaged under CLSM to detect the ecto-CRT exposure, and the excitation of Alexa Fluor 647 was 647 nm and emission was at 688 ± 20 nm. The calreticulin exposure was further detected by flow cytometry. After different treatments, 4T1 cancer cells were collected and washed with FACS buffer. Afterward, the cells were stained with an anti-calreticulin specific antibody (1:200, Abcam, Rabbit mAb, #ab92516) at 4 °C for 30 min. Then, they were washed with FACS buffer and incubated with secondary donkey anti-rabbit IgG H&L (Alexa Fluor® 647) (#ab150075, abcam, 1:500) at 4 °C for 30 min. Sytox Green (S7020, ThermoFisher scientific, USA) was used as the permeabilization marker. Finally, flow cytometry was used to detect the expression of calreticulin. Only non-permeabilized cells were gated for the data analysis. To detect the extracellular HMGB1 and ATP, the 4T1 cancer cells were seeded in 6-well plate at the density of $1.5 \times 10^5$ cells/mL. Afterwards, the 4T1 cancer cells were cultured with culture medium containing M1NPs, PNPs, MNPs or M1-MPNPs (2 μg mL$^{-1}$) at 37 °C for 24 h, followed by irradiation with white light (10 mW/cm$^2$) for 3 min. At 12 h after light stimulation, the culture supernatants were centrifuged at 13,500 g at 4 °C for 10 min. HMGB1 release in the supernatant was measured using HMGB1 ELISA assay (E-EL-M0676c, Elabscience, China) according to the manufacturer's instructions, and the level of secreted ATP were quantified using ATP bioluminescent assay kit in accordance with the instructions of the product.

## In vitro BMDCs maturation study

BMDCs were isolated from the tibia and femur of BALB/c female mice (7 weeks old) according to the established methods[68]. RMPI 1640 culture medium supplemented with 10% FBS, 20 ng mL$^{-1}$ of GM-CSF and 10 ng mL$^{-1}$ of IL-4 was used to induce BMDCs differentiation. For in vitro BMDC maturation study, 4T1 cancer cells were seeded onto the upper chamber of transwells and cultivated overnight, and then treated with "PBS", "PNPs", "MNPs + L", "M1-PNPs", "MPNPs + L", "M1-MPNPs" and "M1-MPNPs + L" as described above, respectively. Afterward, BMDCs ($5 \times 10^5$ cells per well) were incubated in the lower transwell compartment for 24 h. Finally, BMDCs were collected and stained by anti-CD11c (1:200, Biolegend, #117306), anti-CD80 (1:200, Biolegend, #104708) and anti-CD86 (1:200, Biolegend, #105012), and detected using flow cytometry.

## In vivo tumor tropism study

MPNPs or M1-MPNPs were intravenously injected into 4T1 tumor-bearing BALB/c mice at an equivalent MTPE-TT dose. The fluorescence imaging of animals was observed by IVIS (Berthold Technologies, NightOWL II LB983, Germany) at 2 h, 8 h, 12 h, 24 h, 36 h, and 48 h post NPs injection. The semiquantitative fluorescence intensity of tumor sites from the in vivo images were also quantified. After 48 h of injection, organs were excised from the sacrificed mice for imaging. Immunofluorescence staining was also used to investigate the tumor accumulation and distribution of MPNPs or M1-MPNPs. The 5 μm thickness of tumor sections were cut and stained with anti-CD34 antibody (1:200, Abcam, Rabbit mAb, #ab81289) and DAPI, and then imaged with CLSM.

## In vivo prophylactic tumor vaccination study

The BALB/c mice were randomly divided into two groups named as "PBS" and "M1-MPNPs + L", respectively. Each group contained five mice. For "M1-MPNPs + L" group, 4T1 cells were incubated with 2.5 μg mL$^{-1}$ of M1-MPNPs at 37 °C for 24 h. Then, the 4T1 cells were irradiated under white light (10 mW cm$^{-2}$) for 5 min. The mice were then immunized on day 0 by subcutaneous injection of $2 \times 10^6$ of "M1-MPNPs + L"-pretreated cells (in 100 μL sterile PBS) into the left flank of mice. 100 μL of sterile PBS was injected into mice in the PBS group. After seven days, the animals were challenged with live 4T1 cancer cells by subcutaneous injection of $1 \times 10^6$ of 4T1 cells into the right flank. The tumor growth and survival rates of mice in the two groups were monitored.

## In vivo anti-tumor study

To set up the 4T1 tumor model, $1 \times 10^6$ of 4T1 murine breast cancer cells were injected subcutaneously into the left back of each BALB/c mouse, and treatment experiments were performed on day 7 after tumor inoculation. The 4T1 tumor-bearing mice (≈80 mm$^3$) were randomized into seven groups ($n = 5$ mice per group), named "PBS", "PNPs", "MNPs + L", "M1NPs", "MPNPs + L", "M1-MPNPs", and "M1-MPNPs + L", which were intravenously injected with 200 μL of PBS, PNPs, MNPs, M1NPs, MPNPs and M1-MPNPs (0.5 mg mL$^{-1}$ based on MTPE-TT or 1 mg mL$^{-1}$ based on PTX-NB) twice on day 0 and day 2, respectively. At 24 h post each injection, the 4T1 tumor-bearing mice of "MNPs + L", "MPNPs + L", and "M1-MPNPs + L" groups were irradiated with white light (0.3 W cm$^{-2}$, 10 min). For in vivo study, the light needs to penetrate the skin tissue and large solid tumor to reach the tumor cells, thus the light with higher power density than in vitro experiments was used. The tumor volumes of the mice receiving various treatments were monitored every other day.

The bilateral 4T1 tumor model was established to evaluate the abscopal effect, wherein 4T1 cells ($1 \times 10^6$ per mouse) were inoculated in the left flank of the BALB/c mice subcutaneously to establish primary tumors for different therapy treatments. Six days later, 4T1 cells ($5 \times 10^5$ per mouse) were inoculated in the right flank of the mice subcutaneously to establish the distant tumor without any treatment for imitating cancer metastasis. Subsequently, the mice with bilateral 4T1 tumors were randomized into 7 groups ($n = 5$ mice per group) as follows: "PBS", "PNPs", "MNPs + L", "M1NPs", "MPNPs + L", "M1-MPNPs", and "M1-MPNPs + L", respectively. On day 0 and day 2, only the primary tumor was exposed to light irradiation (0.3 W/cm$^2$, 10 min) at 24 h post each intravenous administration of different NPs (200 μL, 0.5 mg mL$^{-1}$ based on MTPE-TT or 1 mg mL$^{-1}$ based on PTX-NB). Then, the volumes of primary tumors and distant tumors were monitored.

## Analysis of immune cells

Mechanical grinding was used to prepare single cell suspensions from the lymph nodes and tumors of 4T1 tumor-bearing mice on day 14 for flow cytometric analysis. Subsequently, a centrifuge was used to collect the mixed cell suspension, followed by the removal of red blood cells (RBCs) using modified RBC lysis buffer for 5 min at room temperature, followed by two washes of 1 × PBS. To investigate the proportions of mature DCs, the cell suspensions from lymph nodes of various groups were co-stained with anti-CD86-PE (1:200, Biolegend,

#105012), anti-CD80-APC (1:200, Biolegend, #104708), and anti-CD11c-FITC (1:200, Biolegend, #117306) at room temperature in the dark for 15 min, followed by flow cytometric analysis after washing three times with 1 × PBS. The single cell suspensions collected from tumors with various treatments were co-incubated with anti-CD3-FITC (1:200, Biolegend, #100306) and anti-CD8-APC (1:200, Biolegend, #100712) for the analysis of the level of tumor-infiltrating CD8$^+$ T cells at room temperature for 15 min, respectively, followed by flow cytometric analysis after washing three times with 1 × PBS. For studying the immune memory effect, single cell suspensions from lymph nodes of 4T1 tumor-bearing mice were co-stained with anti-CD44-PE antibodies (1:200, Biolegend, #103008), anti-CD8-APC, anti-CD3-FITC, and anti-CD62L-PE/Cyanine7 (1:200, Biolegend, #104418) for 15 min in dark at room temperature to investigate the effector memory T cells (Tem cells, CD3$^+$CD8$^+$CD44$^+$CD62L$^-$), and final analyses were conducted by flow cytometry after washing three times with 1 × PBS.

In addition to flow cytometry analysis, on day 14, the tumors of 4T1-tumor-bearing mice received various treatments were excised for immunofluorescence staining analysis. In brief, these harvested tumor tissues were fixed using 4% of paraformaldehyde solution at 4 °C for 24 h. After that, the fixed tissues were conducted by programmed dehydration using 20%, 30% and 40% of sucrose solutions for three days, respectively (each sucrose solution for one day). Then, the samples were embedded with optimum cutting temperature (OCT) compound and cut to a thickness of 7 μm with a freezing microtome (Leica CM1950) for further immunofluorescence staining by relevant antibodies as follows. For detecting ecto-CRT, the tumor tissue slices were incubated with the primary antibody (1:200, Abcam, Rabbit mAb, #ab92516) overnight at 4 °C, followed by incubation with Alexa Fluor 647-conjugated secondary antibody at room temperature for another 2 h. The obtained slices were then sealed with DAPI, and visualized using CLSM.

## In vivo biocompatibility evaluation

In vivo toxicity of M1-MPNPs was assessed in healthy BALB/c mice. Briefly, on day 0 and day 3, M1-MPNPs (200 μL, 0.5 mg mL$^{-1}$ based on MTPE-TT) were intravenously injected into the healthy mice, respectively ($n = 3$ mice). The PBS-treated healthy mice were used as control ($n = 3$ mice). On day 10, all the mice were sacrificed and the blood was collected for blood chemistry analyses including red/white blood cell counts and liver/spleen function. In addition, the main organs (heart, liver, spleen, lung and kidney) of PBS- and M1-MPNPs-injected mice were harvested, sliced for hematoxylin and eosin (H&E) staining on day 10.

## Statistical analysis

Quantitative data were presented as mean ± standard deviation (SD). Statistical comparisons were made using one-way ANOVA (for multiple comparisons) or two-tailed Student's $t$ test (between two groups). All statistical analysis was performed using GraphPad Prism 8.0.2. $P < 0.05$ was considered to be statistically significant.

## Reporting summary

Further information on research design is available in the Nature Portfolio Reporting Summary linked to this article.

## Data availability

All data are available within the Article, Supplementary Information or Source Data file. Source data are provided with this paper.

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

## Acknowledgements

This work was financially supported by the NSFC (82172081 (J.Q.), 52103168 (J.Q.) and 82102200 (W.L.)), CAMS Initiative for Innovative Medicine (2021-I2M-1-043 (W.L.)), the Science and Technology Program of Tianjin, China (21JCZDJC00970 (J.Q.) and 22JCYBJC01000 (W.L.)), Shenzhen Key Laboratory of Functional Aggregate Materials (ZDSYS20211021111400001 (B.Z.T.)), the Science Technology Innovation Commission of Shenzhen Municipality (KQTD20210811090142053 (B.Z.T.) and JCYJ20220818103007014 (B.Z.T.)), the Fundamental Research Funds for the Central Universities (2021-RC350-006 (W.L.), 63233052 (J.Q.) and 63231199 (J.Q.)) and the State Key Laboratory of Medicinal Chemical Biology (Grant No. 2022002 (W.L.)), Nankai University.

## Author contributions

W.L., J.Q. and B.Z.T. conceived and designed the study. X.K., Y.Z. and J.S. performed the chemical synthesis, NPs preparation and in vitro experiments. Y.Z. and J.S. performed the in vivo experiments. X.K., Y.Z., J.S., L.W., W.L., J.Q. and B.Z.T. analyzed the data and participated in the discussion. X.K., Y.Z., W.L., J.Q. and B.Z.T. contributed to the writing of this paper. W.L., J.Q. and B.Z.T. supervised and revised the project and final approval of the version to be published.

## Competing interests

The authors declare no competing interests.
