## [Peer Review File · Nature Communications]

A photo-triggered self-accelerated nanoplatform for
multifunctional image-guided combination cancer
immunotherapyREVIEWER COMMENTS

Reviewer #1 (Remarks to the Author): with expertise in cancer immunology, immunogenic cell death

In this manuscript, the authors aim to develop a self-accelerated nanoplatform combining an aggregation-induced emission luminogen (AIEgen) and hypoxia-responsive prodrug for multifunctional image-guided combination immunotherapy. The authors use PDT to induce efficient cell death in cancer cells. Moreover, the team developed the high-performance AIEgen with hypoxia-responsive paclitaxel (PTX) prodrug encapsulated it into nanoparticles and further camouflaged it with a macrophage cell membrane. In this way, the tumor-targeting theranostic agent was built. The efficiency of this agent has been tested in vitro and apparently, it can induce ICD. The authors also validated their findings in several therapeutic tumor models.

In general, the work is very interesting and relevant in the field of nanomaterials, PDT, and ICD. Indeed, in the manuscript, the authors try to develop a very elegant but complex system with very high specificity and therapeutic potential. However, some of their conclusions are not completely confirmed by the firm experimental data. There are several important major and minor comments which needed to be carefully addressed by the authors before the manuscript can be considered for publication in the high-impact journal - Nature Communications.

Major Comments

1. In regard to the fabrication and characterization of M1 macrophage membrane camouflaged-NPs: it is important to confirm that these membranes do not contain residual LPS and/or IFN- γ which are used to generate M1 macrophages. LPS/IFN- γ can easily mask the anti-tumor effects and give bias to the results. The appropriate controls are missing, and this should be experimentally better controlled and validated.

2. Fig. 4G- western blotting: loading controls are missing; should be added. The quantification of the WB should be provided. It is needed to show whether these structures

contain also LPS/IFN- γ ; please justify experimentally.

3. The specificity of M1-MPNPs targeting to the cancer cells should be also examined in regard to how non-cancerous cells, for example, epithelial cells react to M1-MPNPs. It is known that epithelial layers suffer dramatically from chemotherapeutic drugs giving the major complications of chemotherapy.

4. The “don't eat me signals” should be demonstrated on M1-MPNPs.

Of note how the authors can explain that M1-MPNPs are taken up only by cancer cells and not taken up by RAW macrophages. Because according to the authors' explanation, M1-MPNPs have don't eat me signals, and therefore they are not taken by RAW macrophages and in vivo in the liver and spleen. However, it is very well known that these don't eat me signals will also block the uptake by any phagocytes. It is not clear why M1-MPNPs should be taken by cancer cells and not by macrophages or cells of RES. These discrepancies in the results interpretation and discussion sections (e.g., see lines 327-328) should be resolved and experimentally justified.

5. The authors used an MTT assay to quantify cell death. Importantly the MTT assay relies on the analysis of the metabolic processes and could not specifically identify dying/dead cells. Therefore, it is strongly advised to use another method for example, based on sytox dyes with annexin-V or other methods to quantify cell death and to confirm the data.

6. The type of regulated cell death is needed to be determined: apoptosis, necroptosis, ferroptosis, or any other type.

7. The author used methods for the analysis of the DAMPs which are not ideal and accurate. For example, the WB of HMGB1 is not convincing because the loading controls also increase. HMGB1 should be analyzed in the SN by ELISA.

The same is true for ecto-CRT. The ecto-CRT can be only properly analyzed when it is done in parallel with sytox staining to exclude internal staining and it can be done only by flow cytometry.

8. Fig 6d: The authors stated that “the tumors of M1-MPNPs-treated mice exhibited much higher NIR fluorescence signal than that of MPNPs, while the liver and spleen possessed decreased NPs accumulation”. It is not convincing because untreated-control mice are lacking. There is an accumulation of M1-MPNPs in the liver/spleen and the authors should analyze their cytotoxic effects on these organs.

9. One of the golden standards to characterize ICD is to analyze the maturation of bone-marrow-derived dendritic cells and tumor prophylactic vaccination model. The authors should use state-of-the-art methods to justify their conclusions on ICD.

10. Analysis of the photo-triggered self-accelerated M1-MPNPs for combination immunotherapy of tumors in mice is done during a very short period- just 14 days. This period should be at least 30 days or until ethical points are reached.

Mino comments

- The article must be proofread. There are a lot of typos and spelling mistakes.

Reviewer #2 (Remarks to the Author): with expertise in nanomedicine, cancer

This work introduces the self-accelerated nanoplatfroms combining an AIEgen and hypoxia-responsive prodrug. The nanoplatfroms have enhanced fluorescence and PA brightness for image-guided treatment. The nanoplatfroms have multiple characteristics, including light-induced PDT, hypoxia-responsive drug release, PDT accelerated drug release, M1 camouflaged, and tumor targeting. Some of these functions have a synergistic effect to enhance treatment effectiveness. The author characterized each of the functions using routine methods, including in vitro and in vivo tests. The paper is well organized. The language is acceptable with some minor issues, so please proofread again.

1. Some abbreviations are not defined. For example, PTX-NB was not defined. Even though from the content of the manuscript, readers can deduce that it means PTX conjugated with 4-nitrobenzyl chloroformate, I suggest properly defining abbreviations to avoid potential

confusion. Another one is PNP, I know it refers to prodrug, but readers might get confused. Since nature communications cover a very wide range of topics, please make sure the language appeals to a wider audience.

2. "In addition, the generation of carbon dioxide byproduct may also promote the PA amplitude via the generation of microbubbles." Was confirmed in your investigation? For example, did you observe an increase in PA signal after ROS?
3. The caption of Fig should include a few words on the fabrication of m1-MPNP so that the main text (line 228) and the caption are related.
4. Why was the fluorescence signal weaker with M1-MPNPs for RAW264.7 cells in figure 4b?
5. Why did the synergetic ICD induction " $1 + 1 > 2$ " and not " $1 + 1 = 2$ "? Can you explain the specific mechanism that led to this?
6. Can you explain why M1-MPNPs is higher than PNPs in Supplementary Fig. 27?
7. Why were different white light parameters used? For the in vivo test, 0.3 W/cm^2 was used, but for in vitro tests, it was 10 mW/cm^2 . The times for some of the tests were also different.
8. Authors should investigate or at least discuss the membrane integrity of M1 macrophage membrane under different conditions that the nanoparticles may encounter after injection, for example, different pH environments.

Reviewer #3 (Remarks to the Author): with expertise in nanoparticles, AIEgens

Many thanks for the opportunity to review the work presented by Kang et al. The authors reported a photo-triggered self-accelerated nanoplatfrom for multifunctional image-guided combination immunotherapy. Although significant amount of experimental data have been supplied to support their suggestions, there are still some flaws to be addressed before further considerations.

Major points

1. The authors suggest that M1-MPNPs have better tumor targeting ability than that of MPNPs which might due to its tumor-homing navigator to enhance the NPs accumulation. However, M1 macrophage membrane might also have different stability to that of MPNPs, which can affect the final tumor targeting results. Therefore, the serum stability between

these two nanoparticles should be compared.

2. The drug loading content for PTX should be given.

3. The activation PTX prodrug in cells was not soundly supported by the given results. I recommend to perform HPLC analysis on the cell lysis before and after light irradiation.

4. We know that PDT can also give strong ICD, so the adding of PTX is redundant or not? Because I see no significant differences between MNPs+L with MPNPs+L in Supplementary Fig. 33 and 34.

5. I understand that the preparation of PTX prodrug is to give ROS responsive ability of MPNPs. However, direct loading of PTX is more convenient to lower the complexity of the system for future clinical practice. If PTX prodrug has lower cytotoxicity than free PTX, there should be proves because Fig. 5e is not supportive for this suggestion. So, the authors should add additional discussion with experimental supports to give the necessity of using PTX-NB instead of PTX.

6. The safety of the AIE conjugate should be tested.

7. For Fig. 6F, deeper layer showed equal more even more CD34 staining than surface layer, please explain or change photos.

Minor points

1. There are still some typos. For example, Line 31, "... for the strong light absorption ability., ..." there is an additional ".".

2. The language can be further polished.

3. Some references have wrong volume or issue numbers. For example, Ref. 48. Please check throughout the whole manuscript.

Reviewer #4 (Remarks to the Author): with expertise in AIEgens

In this paper, the authors reported a self-accelerated nanoplatfrom combining an aggregation-induced emission luminogen (AIEgen) and hypoxia-responsive prodrug for multifunctional image-guided combination immunotherapy. The near-infrared AIEgen with methoxy substitution simultaneously possesses boosted fluorescence and photoacoustic (PA) brightness for the strong light absorption ability as well as amplified type I and type II photodynamic therapy (PDT) properties via enhanced intersystem crossing process. By

formulating the high-performance AIEgen with a hypoxia-responsive paclitaxel (PTX) prodrug into nanoparticles, and further camouflaging with macrophage cell membrane, the tumor-targeting theranostic agent is built. The integration of fluorescence and PA imaging helps to comprehensively delineate tumor site in a sensitive manner, providing accurate guidance for tumor therapy. The light-induced PDT effect could consume the local oxygen and lead to severer hypoxia, accelerating the release of PTX. As a result, the combination of PDT and PTX chemotherapy is capable of efficiently inducing immunogenic cancer cell death, which could not only elicit strong antitumor immunity to suppress the primary tumor, but also inhibit the growth of distant tumor. This work represents a new strategy to develop robust theranostic agents via rational molecular design for boosting antitumor immunotherapy. Therefore, I would like to recommend its publication after the following questions were addressed.

1. The whole manuscript should be carefully checked to avoid typing error. For example, in abstract: "light absorption ability., as well as amplified".
2. In Fig 2c and 2f, the abscissa scales need to be resets for clearly.
3. In the 'Synthetic processes section', the coupling constant J should be supplied.
4. In the 'Results' section, the article mentioned 'Density function theory (DFT) calculation', therefore, please provide the reference of the relevant calculations.
5. The format of some of the references in the text is incorrect, please check and correct it.

Responses to reviewers' comments for the manuscript titled "A photo-triggered self-accelerated nanoplatform for multifunctional image-guided combination immunotherapy".

We sincerely thank the reviewers for their positive feedback and valuable comments concerning our article. These comments and suggestions are all of great importance to improve the quality of our article. The point-by-point responses to those comments are presented as below and all changes to the manuscript are highlighted by using red colored text. We hope that the reviewers are satisfied with these changes and the manuscript will be accepted for publication in Nature Communications.

Responses to the Comments and Suggestions of Reviewer #1

Reviewer #1 (Remarks to the Author): with expertise in cancer immunology, immunogenic cell death

In this manuscript, the authors aim to develop a self-accelerated nanoplatform combining an aggregation-induced emission luminogen (AIEgen) and hypoxia-responsive prodrug for multifunctional image-guided combination immunotherapy. The authors use PDT to induce efficient cell death in cancer cells. Moreover, the team developed the high-performance AIEgen with hypoxia-responsive paclitaxel (PTX) prodrug encapsulated it into nanoparticles and further camouflaged it with a macrophage cell membrane. In this way, the tumor-targeting theranostic agent was built. The efficiency of this agent has been tested in vitro and apparently, it can induce ICD. The authors also validated their findings in several therapeutic tumor models.

In general, the work is very interesting and relevant in the field of nanomaterials, PDT, and ICD. Indeed, in the manuscript, the authors try to develop a very elegant but complex system with very high specificity and therapeutic potential. However, some of their conclusions are not completely confirmed by the firm experimental data. There are several important major and minor comments which needed to be carefully addressed by the authors before the manuscript can be considered for publication in the high-impact journal - Nature Communications.

Major Comments

1. In regard to the fabrication and characterization of M1 macrophage membrane camouflaged-NPs: it is important to confirm that these membranes do not contain residual LPS and/or IFN- γ which are used to generate M1 macrophages. LPS/IFN- γ can easily mask the anti-tumor effects and give bias to the results. The appropriate controls are missing, and this should be experimentally better controlled and validated.

Response: We sincerely thank the reviewer for your positive feedback and valuable suggestions to our work. In this work, after stimulation with LPS (100 ng/mL) and IFN- γ (50 ng/mL), the resulting M1 macrophages were washed with PBS three times and then collected via centrifugation, and the further extraction of cell membrane also involved multiple washing and centrifuging steps, so the LPS and IFN- γ should be removed. To further investigate whether residual LPS existed after these purification processes, the M1 macrophage membrane was analyzed using LPS ELISA kit. And for IFN- γ analysis, the M1 macrophages were stained with anti-IFN- γ antibodies and then analyzed by flow cytometry. As shown in Fig. 1, the M1 macrophage membrane did not show any increased level of LPS and IFN- γ , and nearly negligible amounts of LPS and IFN- γ were detected from both cell membrane, indicating the LPS and IFN- γ stimulation did not cause residual LPS and/or IFN- γ on M1 macrophage membrane. We have added the results to the revised

Supplementary information (new Supplementary Fig. 25) and discussed them in the revised manuscript (line 1-5, page 9).

Fig. 1. (a) The concentrations of LPS on M0 and M1 macrophage membrane detected by LPS ELISA kit. (b) Flow cytometry analyses of the IFN- γ on M0 and M1 macrophages.

2. Fig. 4G- western blotting: loading controls are missing; should be added. The quantification of the WB should be provided. It is needed to show whether these structures contain also LPS/IFN- γ ; please justify experimentally.

Response: Thanks for the reviewer's comments. For western blotting analysis of the membrane proteins, Na,K-ATPase was used as the loading control, and the new results were displayed in the revised Fig. 4g. It was evident that both the macrophage membrane and macrophage membrane-coated NPs maintained the characteristic membrane proteins (CD86 and iNOS) from macrophage cells. The quantification of the western blotting was also provided in the revised manuscript (new Supplementary Fig. 27). For LPS/IFN- γ analysis, please see the response for Question 1.

Fig. 2. Representative western blots of CD86 and iNOS expression in different formulations.

Fig. 3. Quantification analyses of CD86 and iNOS expression in different formulations from western blots.

3. The specificity of M1-MPNPs targeting to the cancer cells should be also examined in regard to how not cancerous cells, for example, epithelial cells are react to M1-MPNPs. It is known that epithelial layers suffer dramatically from chemotherapeutic drugs giving the major complications of chemotherapy.

Response: Thanks for the reviewer’s nice suggestion. To evaluate the targeting specificity, M1-MPNPs were incubated with human breast epithelial cells (MCF-10A), human renal proximal tubule epithelial cells (HK-2) and 4T1 cancer cells, respectively. The fluorescence signal of M1-MPNPs in cells was assessed by fluorescence microscopy at 4 h after incubation. As indicated in Fig. 4, much brighter fluorescence signal was observed in 4T1 cancer cells compared with that of MCF-10A and HK-2 epithelial cells, suggesting that M1-MPNPs had better targeting ability toward cancer cells over healthy epithelial cells. The results have been added (new Supplementary Fig. 29) and discussed in the revised manuscript (line 7-11, page 10).

Fig. 4. Representative CLSM images of MCF-10A, HK-2 epithelial cells and 4T1 tumor cells after incubation with M1-MPNPs for 4 h.

4. The “don’t eat me signals” should be demonstrated on MI-MPNPs.

Of note how the authors can explain that MI-MPNPs are taken up only by cancer cells and not taken up by RAW macrophages. Because according to the authors’ explanation, MI-MPNPs have don’t eat me signals, and therefore they are not taken by RAW macrophages and in vivo in the liver and spleen. However, it is very well known that these don’t eat me signals will also block the uptake by any phagocytes. It is not clear why MI-MPNPs should be taken by cancer cells and not by macrophages or cells of RES. These discrepancies in the results interpretation and discussion sections (e.g., see lines 327-328) should be resolved and experimentally justified.

Response: In this work, we found that the macrophage membrane-coated NPs showed increased uptake in 4T1 cancer cells but lower internalization in RAW 264.7 macrophages compared with the uncoated NPs. This finding was consistent with previous literatures (Nat. Commun. 2020, 11, 2622; ACS Nano 2016, 10, 7738–7748; Biomaterials, 2020, 255, 120159; Adv. Healthc. Mater. 2022, 11, e2101788), which also reported that the macrophage membrane coating could not only enhance the tumor-targeting ability of NPs but also decrease their uptake by macrophage. According to previous researches, this was probably attributable to the inherited proteins from macrophage membrane, such as CD47, which could prevent undesirable macrophage-mediated phagocytosis by binding SIRP α expressed on macrophages (Molecular Therapy: Nucleic Acids 2022, 27, 349; Theranostics 2021, 11, 164-180; Adv. Healthc. Mater. 2022, 11, e2101788; J. Nanobiotechnol. 2022, 20, 542). The delayed clearance of NPs by reticuloendothelial system (RES) prolonged their blood circulation, which allowed NPs to accumulate effectively in the tumor site. In addition, it has been reported that the existing surface proteins on macrophage cell membrane, such as $\alpha 4$ and $\beta 1$ integrins, could actively bind to the vascular cell adhesion molecule-1 (VCAM-1) on cancer cells, endowing the macrophage membrane with tumor-targeting ability (Clin. Cancer

Res. 2012, 18, 5520–5525; Cancer Cell, 2011, 20, 538–549; ACS Nano 2016, 10, 7738–7748; J. Nanobiotechnol. 2022, 20, 542; J. Nanobiotechnol. 2020, 18, 92; ACS Nano 2021, 15, 20377–2039). According to these literatures, we have carried out western blotting analysis, and we also observed the expression of these related proteins, including CD47, $\alpha 4$ and $\beta 1$ integrins on M1 macrophage membrane and M1-MPNPs (Fig. 5). Thus CD47 protein probably help to avoid phagocytosis by phagocytes, and the tumor-targeting $\alpha 4$ and $\beta 1$ integrins might promote the binding of NPs with tumor site. In further, we will conduct more deep study to investigate the mechanism behind the differences in cellular uptake. According to reviewer’s suggestion, we have removed the “don’t eat me” sentence and added the above discussions in corresponding part of the revised manuscript (line 11-20, page 10).

Fig. 5. Western blots of CD47, $\alpha 4$ and $\beta 1$ expression on M1 macrophage membrane and M1-MPNPs.

5. The authors used an MTT assay to quantify cell death. Importantly the MTT assay relies on the analysis of the metabolic processes and could not specifically identify dying/dead cells. Therefore, it is strongly advised to use another method for example, based on sytox dyes with annexin-V or other methods to quantify cell death and to confirm the data.

Response: According to the reviewer’s suggestion, we further evaluated the cell death using annexin-V and Sytox Green co-staining method. Annexin V staining is commonly used for detecting phosphatidylserine exposure on apoptotic cells, and sytox dye can permeate dead cells to stain them with intense fluorescence by binding to cellular nucleic acids. After treating 4T1 cells with different formulations, the cells were stained with Annexin V-mCherry and Sytox Green, and then subjected to flow cytometry analysis. As demonstrated in Fig. 6, the extent of Annexin V⁺ and Sytox Green⁺ cells greatly increased after the treatment of M1-MPNPs under light irradiation, indicating that “M1-MPNPs + L” treatment could induce high level of cell apoptosis and death. The results were consistent with that of MTT assay, indicating potent tumor cell-killing ability of M1-MPNPs under light irradiation. The new results have been added (new Supplementary Fig. 33) and discussed in the revised manuscript (line 3-10, page 11).

Fig. 6. Representative flow cytometry results and quantitative analyses of the population of annexin-V and Sytox Green co-staining 4T1 cells after various treatments.

6. The type of regulated cell death is needed to be determined: apoptosis, necroptosis, ferroptosis, or any other type.

Response: According to the reviewer's suggestion, we have studied the types of regulated cell death by using different kinds of cell death inhibitors. For example, Liproxstatin-1 (Lip-1), z-VAD-fmk, and Necrostatin 1s (Nec1s) were used as the inhibitors of ferroptosis, apoptosis and necroptosis, respectively (Nat. Commun. 2022, 13, 3676; Cell Death Discov. 2022, 8, 501). As shown in Fig. 7, the cell death caused by "M1-MPNPs + L" was only slightly inhibited by Lip-1 treatment. However, the addition of z-VAD-fmk or Nec1s significantly inhibited the cell death, suggesting that apoptosis and necroptosis were the two main forms of cell death involved in the treatment used in this work. The results were also consistent with previous literatures, which reported that PDT treatment could induce several cell death modalities, including apoptosis, necroptosis, pyroptosis, and so on (Cell Death Dis. 2022, 13, 455). In further, we will conduct in-depth investigation to study whether other cell death pathways are involved. According to reviewer's suggestion, we have added the results (new Supplementary Fig. 34) and more discussions in the revised manuscript (line 10-16, page 11).

Fig. 7. Cell viability of the 4T1 tumor cells with the treatment of “M1-MPNPs + L” after adding different kinds of cell death inhibitors. No “M1-MPNPs + L” treatment and inhibitor were applied for the PBS group, and no inhibitor was added in the Control group. Error bars: mean \pm SD ($n = 3$). *** $p < 0.001$ using one-way ANOVA.

7. The author used methods for the analysis of the DAMPs which are not ideal and accurate.

For example, the WB of HMGB1 is not convincing because the loading controls also increase.

HMGB1 should be analyzed in the SN by ELISA.

The same is try for ecto-CRT. The ecto-CRT can be only properly analyzed when it is done in parallel with sytox staining to exclude internal staining and it can be done only by flow cytometry.

Response: Based on the reviewer’s comments, the HMGB1 released in the cell supernatant was analyzed using ELISA assay according to the manufacturer’s instructions. The concentrations of HMGB1 in different treatment groups were presented in Fig. 8, and we found an obvious release of HMGB1 from 4T1 cells with “M1-MPNPs + L” treatment.

The calreticulin exposure was also analyzed by flow cytometry according to the reviewer’s suggestion. After the treatment of 4T1 cells with different formulations, the cells were washed

with FACS buffer, and stained with anti-calreticulin antibody. In parallel, Sytox Green was used as the permeabilization marker. The cells were then subjected to flow cytometry analysis, and the histograms represented the fluorescence intensity of calreticulin detected in non-permeabilized (Sytox Green⁻) cells. Compared with other groups, the cells treated with “M1-MPNPs + L” again showed the highest level of calreticulin exposure. The results have been added (new Fig. 5g,5h) and discussed in the revised manuscript (line 29-31, page 11).

Fig. 8. Analysis of HMGB1 concentrations in the supernatant of 4T1 tumor cells after different treatments by ELISA assay. Error bars: mean \pm SD ($n = 3$). *** $p < 0.001$ in comparison with “M1-MPNPs + L” group via one-way ANOVA.

Fig. 9. (a) Gating strategy for calreticulin exposure experiments. (b) Quantitative analyses of the calreticulin exposure in 4T1 tumor cells after various treatments. Error bars: mean \pm SD ($n = 3$). *** $p < 0.001$ in comparison with “M1-MPNPs + L” group via one-way ANOVA.

8. Fig 6d: The authors stated that “the tumors of M1-MPNPs-treated mice exhibited much higher NIR fluorescence signal than that of MPNPs, while the liver and spleen possessed decreased NPs accumulation”. It is not convincing because untreated-control mice are lacking. There is an accumulation of M1-MPNPs in the liver/spleen and the authors should analyze their cytotoxic effects on these organs.

Response: As the reviewer suggested, we have measured the fluorescence images of the untreated mice (the tumor-bearing mice without any treatment). And nearly no fluorescence signal was observed from the tumor and major organs of the untreated mice under the same imaging condition (Fig. 10).

To analyze the toxic effect of M1-MPNPs on spleen, the spleens from the healthy mice with the treatment of PBS or M1-MPNPs were collected and their weight was measured, which showed no noticeable difference (Fig. 11). In addition, the major organs (including heart, liver, spleen, lung, and kidney) were harvested and subjected to H&E staining to determine the histopathology of tissue. As illustrated in the H&E staining images (Fig. 12), the NP-treated groups did not show any noticeable pathological abnormalities. Furthermore, the blood was drawn to analyze the important hematological indicators and biochemical parameters of mice. The hematology parameters and hepatorenal indicators of mice treated with M1-MPNPs were similar to those of PBS-treated mice, and were all within the normal ranges (Fig. 13,14). Taken together, these results suggested that M1-MPNPs had good in vivo biocompatibility and little toxic effect on liver and spleen. The results have been added (new Fig. 5abe, new Supplementary Fig. 46-49) and discussed in the revised manuscript (line 6-11, page 17).

Fig. 10. (a) Representative fluorescence images of 4T1 tumor-bearing mice and (b) corresponding fluorescence intensities in the tumor regions at various time points post i.v. injection of MPNPs or M1-MPNPs, the tumor-bearing mice with no treatment were used as control. Error bars: mean \pm SD ($n = 3$). $*p < 0.05$, $**p < 0.01$ in comparison with “M1-MPNPs + L” group via one-way ANOVA. (c) Representative ex vivo fluorescence image and (d) corresponding fluorescence intensities of major organs and tumors isolated from the mice at 24 h post i.v. injection of MPNPs or M1-MPNPs, the organs from the tumor-bearing mice without any treatment were used as control. $*p < 0.05$, $**p < 0.01$ using one-way ANOVA.

Fig. 11. (a) The photograph and (b) weight of the spleens from the mice with the treatment of PBS or M1-MPNPs. Error bars: mean \pm SD ($n = 3$ mice).

Fig. 12. Representative H&E staining of heart, liver, spleen, lung and kidney of healthy mice with the treatment of PBS or M1-MPNPs. Scale bars: 50 μ m.

Fig. 13. Blood routine indexes (white blood cell count (WBC), lymphocyte (Lymph), monocytes (Mon), granulocyte (Gran), platelet distribution width (PDW), Lymphocyte ratio (LYR), red blood cell count (RBC), plateletcrit (PCT), hemoglobin (HGB), hematocrit (HCT), mean corpuscular volume (MCV), mean corpuscular hemoglobin (MCH), mean corpuscular concentration (MCHC),

red blood cell distribution width (RDW), platelets (PLT), mean platelet volume (MPV)) of the mice with the treatment of PBS or M1-MPNPs. Error bars: mean \pm SD ($n = 3$ mice).

Fig. 14. Liver and renal function indexes (alanine transaminase (ALT), aspartate transaminase (AST), γ -glutamyl transpeptidase (γ -GT), albumin (ALB), total bile acid (TBA), creatinine (CREA), and uric acid (UA)) of the mice with the treatment of PBS or M1-MPNPs. Error bars: mean \pm SD ($n = 3$ mice).

9. One of the golden standards to characterize ICD is to analyze the maturation of bone-marrow-derived dendritic cells and tumor prophylactic vaccination model. The authors should use state-of-the-art methods to justify their conclusions on ICD.

Response: We totally agree with the reviewer, and according to this constructive feedback, we conducted the bone-marrow-derived dendritic cells (BMDCs) maturation and tumor prophylactic vaccination model experiments to further validate the ICD effect induced by our nanoplatform. Briefly, BMDCs were isolated from the tibia and femur of BALB/c female mice (7 weeks old) as described previously (Nat. Commun. 2022, 13, 3676; Adv. Funct. Mater. 2021, 31, 2010637). Then 4T1 cancer cells were seeded onto the upper chamber of transwells, and treated with “PBS”, “PNPs”, “MNPs + L”, “M1-PNPs”, “MPNPs + L”, “M1-MPNPs” and “M1-MPNPs + L”, respectively. Afterward, BMDCs were incubated in the lower transwell compartment. Finally, BMDCs were collected and stained by anti-CD11c, anti-CD80 and anti-CD86, and detected using flow cytometer. The co-incubation of BMDCs with 4T1 cancer cells pretreated with “M1-MPNPs + L” caused increased expression levels of CD80 and CD86, suggesting strong DC maturation (Fig. 15).

In addition, the tumor prophylactic vaccination model was also performed. As shown in Fig. 16, the 4T1 tumor cells pretreated with M1-MPNPs + light irradiation were subcutaneously injected into the left flank of mice on day 0 to establish anti-tumor immunity. The mice injected with PBS were subjected as control. On day 7, the mice were challenged with tumor cells by subcutaneous injection of live 4T1 tumor cells into the right flank of mice. The tumor growth on the right site was monitored every two days. The tumor growth curves and survival rate of mice were presented in Fig. 16. Prophylactic vaccination with 4T1 cells killed by M1-MPNPs + light could protect mice against the subsequent tumor cell challenge, and the tumor growth was greatly inhibited compared with the PBS group. The results have been added (new Fig. 5j,k, new Supplementary Fig. 37,39) and discussed in the revised manuscript (line 9-18, page 12; line 25-30, page 13, line 1-4, page 14).

Fig. 15. (a) Schematic illustration of the measurement of BMDCs maturation. The illustration was created with BioRender.com. (b) Representative flow cytometry results and (c) quantitative analysis of the population of BMDCs maturation ($CD11c^+CD80^+CD86^+$) after various treatments. Error bars: mean \pm SD ($n = 3$). *** $p < 0.001$ in comparison with "M1-MPNPs + L" group via one-way ANOVA.

Fig. 16. (a) Schematic illustration of in vivo prophylactic vaccination model experiment schedule. The illustration was created with BioRender.com. (b) The volume of the right tumor from mice immunized with PBS or M1-MPNPs + light-treated cancer cells after the challenge with live cancer cells. Error bars: mean \pm SD ($n = 5$ mice). *** $p < 0.001$ using Student's t-test. (c) Survival curves of mice immunized with PBS or M1-MPNPs + light-treated cancer cells after the challenge with live cancer cells ($n = 5$ mice).

10. Analysis of the photo-triggered self-accelerated M1-MPNPs for combination immunotherapy of tumors in mice is done during a very short period- just 14 days. This period should be at least 30 days or until ethical points are reached.

Response: According to the reviewer's suggestion, the therapeutic effect of M1-MPNPs on tumor was assessed for a long period of time. The body weight change, growth curves of both primary

and distant tumors of mice were monitored for 30 days, and the data were presented in Fig. 17,18. These results also demonstrated that “M1-MPNPs + L” treatment could more efficiently inhibit the growth of both primary and distant tumors of tumor-bearing mice in comparison with other groups. The results have been added to the revised manuscript (new Fig. 8b-e, Supplementary Fig. 42).

Fig. 17. Plots of (a) body weight, (b) primary tumor volume and (c) individual tumor growth curve of primary tumor of the bilateral 4T1 tumor-bearing mice post various treatments. Error bars: mean \pm SD ($n = 5$).

Fig. 18. Plots of (a) distant tumor volume and (b) individual tumor growth curve of distant tumor of the bilateral 4T1 tumor-bearing mice post various treatments. Error bars: mean \pm SD ($n = 5$).

Mino comments

- The article must be proofread. There are a lot of typos and spelling mistakes.

Response: We sincerely thank the reviewer for pointing out this. We have carefully proof-read the manuscript to minimize typographical and grammatical errors. We believe that the manuscript has been greatly improved.

Responses to the Comments and Suggestions of Reviewer #2

Reviewer #2 (Remarks to the Author): with expertise in nanomedicine, cancer

This work introduces the self-accelerated nanoplatfoms combining an AIEgen and hypoxia-responsive prodrug. The nanoplatfoms have enhanced fluorescence and PA brightness for image-guided treatment. The nanoplatfoms have multiple characteristics, including light-induced PDT, hypoxia-responsive drug release, PDT accelerated drug release, MI camouflaged, and tumor targeting. Some of these functions have a synergistic effect to enhance treatment effectiveness. The author characterized each of the functions using routine methods, including in vitro and in vivo tests. The paper is well organized. The language is acceptable with some minor issues, so please proofread again.

1. Some abbreviations are not defined. For example, PTX-NB was not defined. Even though from the content of the manuscript, readers can deduce that it means PTX conjugated with 4-nitrobenzyl chloroformate, I suggest properly defining abbreviations to avoid potential confusion. Another one is PNP, I know it refers to prodrug, but readers might get confused. Since nature communications cover a very wide range of topics, please make sure the language appeals to a wider audience.

Response: We sincerely thank the reviewer for your careful reading and positive feedback to our work. According to the reviewer's nice suggestions, we have carefully proof-read the manuscript to minimize typographical and grammatical errors, and defined all abbreviations the first time they appear in the revised manuscript.

2. “In addition, the generation of carbon dioxide byproduct may also promote the PA amplitude via the generation of microbubbles.” Was confirmed in your investigation? For example, did you observe an increase in PA signal after ROS?

Response: The statement was originated from the previous reports that the in situ-generated gas bubbles could enhance PA signal through the microbubble-triggered cavitation effect (Angew. Chem. Int. Ed. 2018, 57, 10309; ACS Nano 2018, 12, 392; Adv. Mater. 2022, 34, 2108348). However, we did not test this effect directly in our study. To avoid any possible confusion, we have deleted the sentence in the revised manuscript.

3. The caption of Fig should include a few words on the fabrication of m1-MPNP so that the main text (line 228) and the caption are related.

Response: We thank for the reviewer’s nice suggestion. In the revised manuscript, we have added some description about the fabrication of M1-MPNP in the caption of Fig. 1.

4. Why was the fluorescence signal weaker with M1-MPNPs for RAW264.7 cells in figure 4b?

Response: In this work, we found that the macrophage membrane-coated NPs showed increased uptake in 4T1 cancer cells but lower internalization in RAW 264.7 macrophages compared with the uncoated NPs. This finding was consistent with previous literatures (Biomaterials, 2020, 255, 120159; Nat. Commun. 2020, 11, 2622; Adv. Healthc. Mater. 2022, 11, e2101788), which also reported that the macrophage membrane coating could not only enhance the tumor-targeting ability of NPs but also decrease their uptake by macrophage. According to previous researches, this was

probably attributable to the inherited proteins from macrophage membrane, such as CD47, which could prevent undesirable macrophage-mediated phagocytosis by binding SIRP α expressed on macrophages (Molecular Therapy: Nucleic Acids 2022, 27, 349; Theranostics 2021, 11, 164-180; Adv. Healthc. Mater. 2022, 11, e2101788; J. Nanobiotechnol. 2022, 20, 542). To verify this, we conducted western blotting analysis, and we also observed the expression of CD47 on M1 macrophage membrane and M1-MPNPs (Fig. 1), which probably helped to avoid phagocytosis by phagocytes. We have added some discussions about the low uptake of M1-MPNPs by RAW cells in corresponding part of the revised manuscript (line 11-20, page 10).

Fig. 1. Western blots of CD47 expression on M1 macrophage membrane and M1-MPNPs.

5. Why did the synergetic ICD induction “1 + 1 > 2” and not “1 + 1 = 2”? Can you explain the specific mechanism that led to this?

Response: In this work, to reduce systemic side effect, the hypoxia-responsive prodrug (PTX-NB) was used, which could release free PTX drug at the hypoxic tumor site to induce the ICD of cancer cells. On the other hand, the AIEgen-based PDT effect could not only act as another inducer to active ICD but also consume the local oxygen to lead to more severely hypoxic tumor microenvironment, which accelerated the release of more free PTX drugs in tumor. As a result,

the “1 + 1 > 2” synergetic ICD effect would be expected. As a response to reviewer’s comment, we have discussed this in the revised manuscript (line 3-4, page 12).

6. Can you explain why M1-MPNPs is higher than PNPs in Supplementary Fig. 27?

Response: Supplementary Fig. 27 suggested that the proportion of CRT exposure on M1-MPNPs-treated tumor cells was 1.3 folder higher than that of PNPs-treated tumor cells. This was probably due to the enhanced tumor cell-targeting ability mediated by the M1 macrophage membrane coating. Thus, as compared with PNPs, a higher amount of M1-MPNPs could be uptaken by 4T1 tumor cells, and more PTX drug would be partially released to induce ICD in the hypoxia microenvironment. Therefore, the M1-MPNPs-treated 4T1 tumor cells exhibited higher proportion of CRT exposure than the PNPs group. We have discussed this in the revised Supplementary information.

7. Why were different white light parameters used? For the in vivo test, 0.3 W/cm² was used, but for in vitro tests, it was 10 mW/cm². The times for some of the tests were also different.

Response: For in vitro experiments, the light source could readily reach the NPs solution or cell media, thus the light with relatively low power was enough. However, for in vivo study, the light needs to penetrate the skin tissue and large solid tumor to reach the tumor cells, so the light with high power density was required to achieve effective treatment outcome. Many literatures have also reported the using of light with different power densities or irradiation time for in vitro and in vivo experiments, such as J. Am. Chem. Soc. 2023, 145, 4081 and Nat. Commun. 2020, 11, 357.

According to the reviewer's comment, we have explained this in the revised manuscript (line 25-27, page 22).

8. Authors should investigate or at least discuss the membrane integrity of M1 macrophage membrane under different conditions that the nanoparticles may encounter after injection, for example, different pH environments.

Response: We thank for the reviewer's suggestion. Considering the weak acidic environment of tumor (pH around 6.5-7.0) (Nat. Rev. Drug Discov. 2011, 10, 767–777; J. Control. Release 2015, 219, 205-214), we investigated the stability of the membrane-coated M1-MPNPs at pH 6.5. As shown in Fig. 2, nearly no diameter changes were observed when M1-MPNPs were suspended in PBS at pH 7.4 and pH 6.5, indicating the good colloidal stability at both pHs. The result was consistent with previous studies (Procedia Chemistry 2010, 2, 26–33), which also indicated that the cell membrane-based coatings were stable at pH 6.5. In the future, we will conduct in-depth investigations to study the stability and integrity of membrane coating in more complicated physiological conditions and to evaluate how the integrity of membrane affect the tumor targeting ability. According to the reviewer's comments, we have added the pH stability results (Supplementary Fig. 28) and more discussions in corresponding part of the revised manuscript (line 17-22, page 9).

Fig. 2. The colloidal stability of M1-MPNPs in different pHs measured by DLS. Error bars: mean \pm SD ($n = 3$).

Responses to the Comments and Suggestions of Reviewer #3

Reviewer #3 (Remarks to the Author): with expertise in nanoparticles, AIEgens

Many thanks for the opportunity to review the work presented by Kang et al. The authors reported a photo-triggered self-accelerated nanopatform for multifunctional image-guided combination immunotherapy. Although significant amount of experimental data have been supplied to support their suggestions, there are still some flaws to be addressed before further considerations.

Major points

1. The authors suggest that M1-MPNPs have better tumor targeting ability than that of MPNPs which might due to its tumor-homing navigator to enhance the NPs accumulation. However, M1 macrophage membrane might also have different stability to that of MPNPs, which can affect the final tumor targeting results. Therefore, the serum stability between these two nanoparticles should be compared.

Response: We sincerely thank the reviewer for your positive feedback and valuable comments to our work. According to the reviewer's suggestion, we have measured and compared the serum stability of MPNPs and M1-MPNPs. MPNPs and M1-MPNPs were suspended in PBS with 10% fetal bovine serum (FBS), and their diameters were monitored over time by DLS. As illustrated in Fig. 1, the hydrodynamic diameter of both MPNPs and M1-MPNPs remained almost unchanged after storage in serum for 4 days. This result indicated that the M1-MPNPs possessed good colloidal stability in the mimicked physiological fluids. The result was in accordance with previous reports, which also demonstrated the good stability of cell membrane-coated NPs in serum (Adv. Mater. 2022, 2206401; Adv. Mater. 2017, 29, 1606209; ACS Nano 2023, 17, 421). The results

have been added (new Supplementary Fig. 28) and discussed in the revised manuscript (line 16-17, page 9).

Fig. 1. The colloidal stability of MPNPs and M1-MPNPs in FBS. Error bars: mean \pm SD ($n = 3$).

2. The drug loading content for PTX should be given.

Response: Based on the reviewer's suggestion, we have studied the loading content of PTX-NB in M1-MPNPs using high-performance liquid chromatography (HPLC). The HPLC analysis indicated that the drug loading content of PTX-NB in M1-MPNPs was about 8.3%. And the results have been discussed in the revised manuscript (line 15-16, page 9).

3. The activation PTX prodrug in cells was not soundly supported by the given results. I recommend to perform HPLC analysis on the cell lysis before and after light irradiation.

Response: According to the reviewer's nice suggestion, we have studied the activation of PTX-NB prodrug in cells using HPLC. M1-MPNPs were incubated with 4T1 cancer cells at 37 °C for

4 h, and then the cells were washed with $1 \times$ PBS and exposed to white light (10 mW/cm^2) for 5 min. Finally, the cell lysates were subjected to HPLC analysis. For the cell lysates from light-treated cells, the peak of PTX-NB at 10.2 min decreased while the peak representing PTX increased when compared to that from the cells without light irradiation (Fig. 2). This result indicated that the PDT effect could accelerate the conversion of PTX-NB to free PTX in cells. We have added the result to the revised manuscript (new Supplementary Fig. 31).

Fig. 2. The analysis of PTX activation by HPLC. The lysates from M1-MPNPs-treated cells before and after white light irradiation were measured by HPLC analysis. The peak of PTX-NB at 10.2 min (b) decreased while the peak representing PTX (a) increased after the cells were irradiated by light.

4. We know that PDT can also give strong ICD, so the adding of PTX is redundant or not? Because I see no significant differences between MNPs+L with MPNPs+L in Supplementary Fig. 33 and 34.

Response: In this work, to achieve potent antitumor outcomes, PDT and hypoxia-responsive prodrug were combined for self-accelerated ICD induction and synergistic immunotherapy. To

address the reviewer’s concern, we quantified the ecto-CRT expression results in Supplementary Fig. 33 and the tumor-infiltrating CD8⁺ T cell results in Supplementary Fig. 34. And the quantitative analysis indicated that the “MPNPs + L” treatment could significantly increase the expression levels of ecto-CRT as well as the proportions of tumor-infiltrating CD8⁺ T cells in primary tumors when compared to “MNPs + L”. In addition, the quantitative data of the percentages of CD8⁺ T cells in distant tumors (Fig. 8j) also indicated that the combination of PDT and PTX-NB prodrug could lead to superior anti-tumor immunity than PDT alone. According to the reviewer’s concern, we have put the quantitative analysis results in the revised Supplementary information (new Supplementary Fig. 44).

Fig. 3. Quantitative analysis results of (a) ecto-CRT expression and (b) the proportions of tumor-infiltrating CD8⁺ T cells in primary tumors collected from bilateral 4T1 tumor-bearing mice after various treatments. Error bars: mean ± SD ($n = 3$). *** $p < 0.001$ using one-way ANOVA.

5. I understand that the preparation of PTX prodrug is to give ROS responsive ability of MPNPs. However, direct loading of PTX is more convenient to lower the complexity of the system for future

clinical practice. If PTX prodrug has lower cytotoxicity than free PTX, there should be proves because Fig. 5e is not supportive for this suggestion. So, the authors should add additional discussion with experimental supports to give the necessity of using PTX-NB instead of PTX.

Response: In this work, the hypoxia-responsive PTX-NB prodrug was used to reduce the side effect that may be caused by the acculturation of free chemodrugs in healthy organs. Chemotherapy is usually associated with severe systemic toxicity (Nat. Rev. Clin. Oncol. 2016, 13, 92–105; Front. Pharmacol. 2018, 9, 245). Despite that nanodrug delivery system could improve the accumulation of chemotherapeutic drug at tumor site, they still unavoidably enter other normal organs. The PTX-NB prodrugs are able to be selectively activated in the hypoxia tumor microenvironment to release free PTX drug, which would further increase the safety of chemodrug. To demonstrate that the PTX-NB prodrug-based nanoplatfoms had reduced side effect, M1-MPNPs were incubated with the normal breast epithelial cell line, MCF-10A. For control, the free PTX-based NPs (M1-MFNPs) were prepared by using free PTX to replace the PTX-NB prodrug, and then incubated with MCF-10A cells. The MTT assay indicated that M1-MPNPs showed obviously reduced cytotoxicity toward MCF-10A cells than M1-MFNPs (Fig. 4). The results indicated that the PTX prodrug-based nanoplatfom could reduce the toxic side effect caused by PTX free drug. According to the reviewer's comments, we have added these results (new Supplementary Fig. 35) and more discussions in corresponding part of the revised manuscript (line 16-20, page 11).

Fig. 4. MTT assay of MCF-10A cells with the treatment of different concentrations of M1-MFNPs or M1-MPNPs. Error bars: mean \pm SD ($n = 3$). * $p < 0.05$, ** $p < 0.01$ using Student's t-test.

6. The safety of the AIE conjugate should be tested.

Response: According to the reviewer's suggestion, we further evaluated the in vivo safety profile of M1-MPNPs in detail. After the intravenous administration of M1-MPNPs or PBS into healthy mice, the blood of mice was drawn to analyze the important hematological indicators and biochemical parameters. The hematology parameters for mice that received M1-MPNPs were not significantly different from those of the PBS-injected mice (Fig. 5). Furthermore, multiple serum biochemical markers, including alanine transaminase (ALT), aspartate transaminase (AST), γ -glutamyl transpeptidase (γ -GT), albumin (ALB), total bile acid (TBA), creatinine (CREA), and uric acid (UA) were measured to detect the liver and kidney functions. And all these hepatorenal indicators were within the normal ranges (Fig. 6). In our previous data, the major organs (including heart, liver, spleen, lung, and kidney) were harvested and subjected to H&E staining to determine the histopathology of tissues. Similar to PBS-injected mice, the organs of NP-treated groups did not show any noticeable pathological abnormalities. Taken together, these results collectively

suggested the good in vivo biocompatibility of M1-MPNPs. The results have been added (new Supplementary Fig. 48,49) and discussed in the revised manuscript (line 6-11, page 17).

Fig. 5. Blood routine indexes (white blood cell count (WBC), lymphocyte (Lymph), monocytes (Mon), granulocyte (Gran), platelet distribution width (PDW), Lymphocyte ratio (LYR), red blood cell count (RBC), plateletcrit (PCT), hemoglobin (HGB), hematocrit (HCT), mean corpuscular volume (MCV), mean corpuscular hemoglobin (MCH), mean corpuscular concentration (MCHC), red blood cell distribution width (RDW), platelets (PLT), and mean platelet volume (MPV)) of the mice with the treatment of PBS or M1-MPNPs. Error bars: mean \pm SD ($n = 3$ mice).

Fig. 6. Blood test parameters included liver and renal function indexes (alanine transaminase (ALT), aspartate transaminase (AST), γ -glutamyl transpeptidase (γ -GT), albumin (ALB), total bile acid (TBA), creatinine (CREA), and uric acid (UA)) of the mice with the treatment of PBS or M1-MPNPs. Error bars: mean \pm SD ($n = 3$ mice).

7. For Fig. 6F, deeper layer showed equal more even more CD34 staining than surface layer, please explain or change photos.

Response: We thank the reviewer for pointing this out. According to the reviewer's suggestion, we have replaced the surface layer photos with another group of images we took previously. And the new results and quantitative analysis were shown in Fig. 7 (below) and new Fig. 6f in the revised manuscript.

Fig. 7. Representative fluorescent images of tumor sections from the mice injected with MPNPs or M1-MPNPs. Blue: DAPI for staining cell nucleus; green: CD34 for staining tumor neovasculature; and red: NPs.

Minor points

1. There are still some typos. For example, Line 31, "... for the strong light absorption ability., ..." there is an additional ".".

Response: We feel sorry for these typos. We have corrected them and also carefully checked the whole manuscript to avoid typographical errors.

2. The language can be further polished.

Response: According the reviewer's nice suggestion, we have carefully proof-read the manuscript to minimize typographical and grammatical errors, and we believe that the manuscript has been greatly improved.

3. Some references have wrong volume or issue numbers. For example, Ref. 48. Please check throughout the whole manuscript.

Response: We thank for the reviewer's suggestion. We have carefully checked and revised the format of all the references.

Responses to the Comments and Suggestions of Reviewer #4

Reviewer #4 (Remarks to the Author): with expertise in AIEgens

In this paper, the authors reported a self-accelerated nanoplatfrom combining an aggregation-induced emission luminogen (AIEgen) and hypoxia-responsive prodrug for multifunctional image-guided combination immunotherapy. The near-infrared AIEgen with methoxy substitution simultaneously possesses boosted fluorescence and photoacoustic (PA) brightness for the strong light absorption ability as well as amplified type I and type II photodynamic therapy (PDT) properties via enhanced intersystem crossing process. By formulating the high-performance AIEgen with a hypoxia-responsive paclitaxel (PTX) prodrug into nanoparticles, and further camouflaging with macrophage cell membrane, the tumor-targeting theranostic agent is built. The integration of fluorescence and PA imaging helps to comprehensively delineate tumor site in a sensitive manner, providing accurate guidance for tumor therapy. The light-induced PDT effect could consume the local oxygen and lead to severer hypoxia, accelerating the release of PTX. As a result, the combination of PDT and PTX chemotherapy is capable of efficiently inducing immunogenic cancer cell death, which could not only elicit strong antitumor immunity to suppress the primary tumor, but also inhibit the growth of distant tumor. This work represents a new strategy to develop robust theranostic agents via rational molecular design for boosting antitumor immunotherapy. Therefore, I would like to recommend its publication after the following questions were addressed.

1. The whole manuscript should be carefully checked to avoid typing error. For example, in abstract: “light absorption ability., as well as amplified”.

Response: We sincerely thank the reviewer for your careful reading and positive feedback to our work. We have carefully checked the whole manuscript and corrected the typing errors in the revised manuscript.

2. In Fig 2c and 2f, the abscissa scales need to be resets for clearly.

Response: We thank the reviewer for the nice suggestion. We have reset the abscissa scales of Fig. 2c and 2f to make them clearer.

3. In the ‘Synthetic processes section’, the coupling constant J should be supplied.

Response: Based on the reviewer’s suggestion, and we have provided the coupling constant J in the “Synthetic processes section” in the revised Supplementary information.

4. In the ‘Results’ section, the article mentioned ‘Density function theory (DFT) calculation’, therefore, please provide the reference of the relevant calculations.

Response: According to the reviewer’s kind suggestion, we have provided the reference for the density function theory (DFT) calculation in the revised manuscript.

5. The format of some of the references in the text is incorrect, please check and correct it.

Response: Thank you for the careful reading. We have checked and corrected the format of all the references.

REVIEWERS' COMMENTS

Reviewer #1 (Remarks to the Author):

The authors addressed most of the comments in a proper way and have provided an adapted manuscript accordingly. Based on all this I would recommend this manuscript for publication in the NC Journal.

Reviewer #2 (Remarks to the Author):

Thank you for the response letter. The responses are detailed and clear. They addressed the comments by adding detailed explanations and/or additional experimental results. I still see some minor language issues, mostly on the use of grammar articles (a, an, the, etc.).

Reviewer #3 (Remarks to the Author):

The authors have addressed all my concerns. The paper can be accepted in its current form.

Reviewer #4 (Remarks to the Author):

This manuscript has been revised according to reviewer's comments and suitable for publication now.

We sincerely thank the reviewers for his/her precious time and recognition of our work. Below are our point-by-point responses to the reviewer's comments.

Responses to the comments and suggestions of Reviewer #1

Reviewer #1 (Remarks to the Author):

The authors addressed most of the comments in a proper way and have provided an adapted manuscript accordingly. Based on all this I would recommend this manuscript for publication in the NC Journal.

Response: We sincerely thank the reviewer for your careful reading and supporting publication of our study.

Responses to the comments and suggestions of Reviewer #2

Reviewer #2 (Remarks to the Author):

Thank you for the response letter. The responses are detailed and clear. They addressed the comments by adding detailed explanations and/or additional experimental results. I still see some minor language issues, mostly on the use of grammar articles (a, an, the, etc.).

Response: We sincerely thank the reviewer for your careful reading and supporting publication of our study. We have carefully checked the language and modified some grammar articles.

Responses to the comments and suggestions of Reviewer #3

Reviewer #3 (Remarks to the Author):

The authors have addressed all my concerns. The paper can be accepted in its current form.

Response: We sincerely thank the reviewer for your careful reading and supporting acceptance of our study.

Responses to the comments and suggestions of Reviewer #4

Reviewer #4 (Remarks to the Author):

This manuscript has been revised according to reviewer's comments and suitable for publication now.

Response: We sincerely thank the reviewer for your careful reading and supporting publication of our study.